# The orphan receptor GPR88 blunts the signaling of opioid receptors and multiple striatal GPCRs

Thibaut Laboute[1†], Jorge Gandía[1†], Lucie P Pellissier[1,2], Yannick Corde[1], Florian Rebeillard[3], Maria Gallo[4], Christophe Gauthier[2], Audrey Léauté[1], Jorge Diaz[3], Anne Poupon[2], Brigitte L Kieffer[5,6], Julie Le Merrer[1,6‡*], Jérôme AJ Becker[1,6‡*]

[1]Deficits of Reward GPCRs and Sociability, Physiologie de la Reproduction et des Comportements, INRA UMR-0085, CNRS UMR-7247, Université de Tours, Inserm, Nouzilly, France; [2]Biology and Bioinformatics of Signalling Systems, Physiologie de la Reproduction et des Comportements, INRA UMR-0085, CNRS UMR-7247, Université de Tours, Nouzilly, France; [3]Cellular Biology and Molecular Pharmacology of central Receptors, Centre de Psychiatrie et Neurosciences, Inserm UMR_S894 - Université Paris Descartes, Sorbonne Paris Cité, Paris, France; [4]Department of Experimental and Health Sciences, Pompeu Fabra University, Barcelona Biomedical Research Park, Barcelona, Spain; [5]Department of Psychiatry, Douglas Mental Health University Institute, McGill University, Montreal, Canada; [6]Institut de Génétique et de Biologie Moléculaire et Cellulaire, CNRS UMR 7104, Inserm U1258, Université de Strasbourg, 1 rue Laurent Fries, Illkirch, France

*For correspondence:
julie.le-merrer@inra.fr (JLM);
jerome.becker@inra.fr (JAJB)

[†]These authors contributed
equally to this work
[‡]These authors also contributed
equally to this work

Competing interests: The
authors declare that no
competing interests exist.

Reviewing editor: Volker
Dötsch, Goethe University,
Germany

**Abstract** GPR88 is an orphan G protein-coupled receptor (GPCR) considered as a promising therapeutic target for neuropsychiatric disorders; its pharmacology, however, remains scarcely understood. Based on our previous report of increased delta opioid receptor activity in *Gpr88* null mice, we investigated the impact of GPR88 co-expression on the signaling of opioid receptors in vitro and revealed that GPR88 inhibits the activation of both their G protein- and β-arrestin-dependent signaling pathways. In *Gpr88* knockout mice, morphine-induced locomotor sensitization, withdrawal and supra-spinal analgesia were facilitated, consistent with a tonic inhibitory action of GPR88 on μOR signaling. We then explored GPR88 interactions with more striatal versus non-neuronal GPCRs, and revealed that GPR88 can decrease the G protein-dependent signaling of most receptors in close proximity, but impedes β-arrestin recruitment by all receptors tested. Our study unravels an unsuspected buffering role of GPR88 expression on GPCR signaling, with intriguing consequences for opioid and striatal functions.

## Introduction

GPR88 is a striatal-enriched orphan G protein-coupled receptor (GPCR) whose expression varies over development in the brain of rodents, monkeys and humans (*Ghate et al., 2007*; *Massart et al., 2009*; *Van Waes et al., 2011*; *Massart et al., 2016*). Gene association studies in humans have uncovered a link between GPR88 function and several psychiatric, neurodevelopmental or neurode-generative disorders, including schizophrenia, bipolar disorder, speech delay and chorea (*Alkufri et al., 2016*; *Del Zompo et al., 2014*). In rodents, transcript levels of *Gpr88* gene were found modified following exposure to various psychoactive drugs, such as mood stabilizers, antide-pressants, methamphetamine, L-DOPA and drugs of abuse (*Conti et al., 2007*; *Ogden et al., 2004*;

*Brandish et al., 2005*; *Le Merrer et al., 2012*; *Becker et al., 2017*). GPR88 thus appears a promising target for the development of innovative treatments for CNS pathologies. In mice, deletion of the *Gpr88* gene alters primarily striatal physiology, striatum-centered brain networks and striatal-dependent behaviors, with notably severe deficits in motor coordination and skill learning, hyperactivity, stereotypies and altered reward-driven behaviors (*Meirsman et al., 2016a*; *Meirsman et al., 2016b*; *Rainwater et al., 2017*; *Ben Hamida et al., 2018*; *Arefin et al., 2017*; *Quintana et al., 2012*). GPR88 function, however, extends beyond striatal-mediated responses, in accordance with extra-striatal GPR88 expression (*Ehrlich et al., 2018a*) and widespread modifications in brain connectivity, gene expression and behavioral responses in *Gpr88* null ($Gpr88^{-/-}$) mice (*Meirsman et al., 2016a*; *Arefin et al., 2017*). Thus, GPR88 has a major influence on brain physiology and controls a vast repertoire of behaviors; the molecular bases of this control, however, remains poorly understood, mostly due to the paucity of pharmacological tools available to manipulate its activity.

At structural level, GPR88 appears as an atypical GPCR. Although considered as part of the class A (rhodopsin) family of GPCRs, GPR88 is distantly related to any well-known receptor of this class (*Surgand et al., 2006*; *Vassilatis et al., 2003*; *Joost and Methner, 2002*; *Kakarala and Jamil, 2014*) and displays an unusually short C-terminus (14 amino acids) and large non-homologous third intracellular loop (67 amino acids). Moreover, based on the sequence of its putative transmembrane binding pocket, GPR88 clusters with class C GPCRs, for which the endogenous ligands bind in the large N-terminus domain (*Surgand et al., 2006*). Intensive efforts in deorphanizing GPR88 have remained vain (*Bi et al., 2015*; *Decker et al., 2017*; *Dzierba et al., 2015*; *Jin et al., 2014*); meanwhile, synthetic chemistry met significant difficulties in identifying potent agonists for this atypical receptor (*Bi et al., 2015*; *Jin et al., 2014*; *Jin et al., 2018*). With only a handful of synthetic ligands available to target GPR88, its pharmacology remains poorly explored. Although this receptor was shown to display constitutive and ligand-induced activity on Gαi/o/z inhibitory protein coupling in heterologous and native cells (*Meirsman et al., 2016a*; *Dzierba et al., 2015*; *Jin et al., 2018*), its ability to recruit β-arrestins and intracellular trafficking have not been investigated yet.

The atypical pharmacology of GPR88 questions the role of an endogenous ligand in its function. Interestingly, it was proposed that orphan GPCRs may influence the signaling of other GPCRs in a ligand-independent manner through hetero-oligomerization, as shown for GPR50 (*Levoye et al., 2006a*; *Levoye et al., 2006b*). Remarkably, we have evidenced in a previous study (*Meirsman et al., 2016a*) an increase in [$^{35}$S]-GTPγS binding of striatal membranes from $Gpr88^{-/-}$ mice in response to stimulation by agonists of the muscarinic and opioid delta (δOR) and mu (µOR) receptors. Moreover, we noticed that behavioral features of these mutants intriguingly oppose several aspects of the phenotype of mice lacking δOR ($Oprd1^{-/-}$) and that pharmacological blockade of δOR ameliorate behavioral deficits in *Gpr88* knockout mice. Together, these data suggest that GPR88 represses the activity of opioid and muscarinic receptors under physiological conditions, with significant consequences on δOR-mediated behavioral responses.

In the present study, we first investigated in vitro whether GPR88 was able to come in close physical proximity to the three opioid receptors, and if its co-expression with these receptors had an influence in their ability to activate G protein and β-arrestin-dependent signaling pathways. Having revealed a significant impact of GPR88 expression on µOR signaling, we then assessed behavioral responses to the µOR agonist morphine in mice lacking the *Gpr88* gene. Intriguingly, the consequences of *Gpr88* deletion on morphine-induced responses were different if not opposed depending on the behavior tested. Considering that morphine-induced responses involve other GPCRs beyond µOR, we extended our in vitro studies to various GPCRs with striatum-enriched versus non-neuronal expression. We unraveled the ability of GPR88, when co-expressed with multiple other GPCRs, to bias their signaling by repressing G protein-dependent activation when closely interacting with them and β-arrestin recruitment independently from physical proximity.

## Results

### GPR88 comes in close physical proximity to opioid receptors and inhibits their signaling

We previously showed that µOR and δOR-mediated G protein signaling is increased in striatal membranes of $Gpr88^{-/-}$ mice (*Meirsman et al., 2016a*), suggesting that, under physiological conditions,

GPR88 represses the activity of these two opioid receptors. Moreover, pharmacological blockade of δOR in *Gpr88* null mice normalizes several aspects of their behavioral phenotype, consistent with excessive δOR activity in these animals. We thus hypothesized that GPR88 can form hetero-oligomers with opioid receptors and then influence their pharmacology.

We first assessed whether GPR88 may come in close physical proximity to opioid receptors using bioluminescence resonance energy transfer (BRET1) saturation assay in HEK293FT cells to explore interactions between Luciferase Rluc8 (RLuc8)-tagged GPR88 and Venus-tagged GPCRs (see examples of converse constructs in *Figure 1—figure supplement 1*). We evidenced that GPR88 displays specific and saturated BRET signals when co-expressed with the three opioid receptors, δOR, κOR and μOR, indicative of close physical proximity (within 10 nm) to them, in addition to itself (*Figure 1A*). These results indicate that GPR88 possibly forms hetero-oligomers with all opioid receptors.

We then tested, in heterologous cells, the consequences of co-expressing GPR88 with opioid receptors on their ability to activate G protein and β-arrestin dependent pathways as well as on their trafficking. Based on our previous results (*Meirsman et al., 2016a*), we first focused on δOR and tested the effects of SNC80 (10 μM) on δOR-mediated signaling in presence of increasing amounts of GPR88 (*Figure 1B*). While δOR cell surface expression (measured using anti-HA antibody, see *Figure 1—figure supplement 2A*) remained constant (*Figure 1—figure supplement 2B*), growing levels of GPR88 expression led to a dose-dependent reduction in SNC80-induced δOR inhibition of cAMP accumulation (Gαi/o-dependent pathway) and β-arrestin2 (β−arr2) recruitment (and similarly β−arr1, *Figure 1—figure supplement 2B*). We then monitored β−arr2 recruitment at δOR in absence (basal) or presence (stimulated) of SNC80, to evaluate the probability of such recruitment in presence or not of GPR88 (30 ng of cDNA transfected). We evidenced that the probability of β−arr2 recruitment at δOR was low under basal conditions and high when δOR was stimulated. GPR88 expression interfered with δOR/β−arr2 interactions under stimulated conditions only. Furthermore, and consistent with δOR internalization being β-arrestin-dependent, we observed diminished loss of interaction between δOR and membrane-expressed Kras, and decreased interaction with early endosome-expressed Rab5 and late endosome-expressed Rab7 (*Figure 1—figure supplement 2B*), indicating decreased SNC80-induced δOR internalization and trafficking in presence of increasing amounts of GPR88. Finally, we used immunochemistry to further explore the consequences of GPR88 on δOR trafficking (*Figure 1C* and *Figure 1—figure supplement 2C*). We first verified membrane expression of GPR88 using a hemagglutinin (HA)-tagged construct (red, *Figure 1C*, Panel a, arrowhead). Similarly, HA-tagged δOR was mostly localized at the plasma membrane when not stimulated (red, Panel b, arrowhead) and SNC80 triggered its complete internalization in vesicular compartments (Panel c, arrowhead). When HA-δOR (red) and GPR88-Venus (green) were co-transfected, they co-localized at plasma membrane under basal and activated conditions (Panels d and e, arrowheads). Upon SNC80 agonist exposure, however, the typical internalization profile of δOR was lost compared to cells expressing δOR only (Panel f, arrowhead versus arrow). Altogether, these data indicate that co-expressing GPR88 with δOR is sufficient to affect G protein and β-arrestin dependent signaling pathways and trafficking of the latter.

We next explored the effects of the selective κOR agonist U50488H (10 μM) on κOR-mediated signaling in presence of increasing amounts of GPR88 (*Figure 1D*). GPR88 expression decreased U50488H-activated κOR pathway to Gαi/o, although in our hands to a lesser extent than for δOR (decrease up to 27.7% of maximal effect versus 61.1% for δOR) and only for the highest dose of GPR88 cDNA, without altering κOR cell surface expression *Figure 1—figure supplement 2D*). GPR88, however, had a significant inhibitory effect on agonist-induced β−arr2 recruitment at κOR (and β−arr1, *Figure 1—figure supplement 2D*). When monitoring the affinity of β−arr2 for κOR, we observed, as for δOR, an increase in such affinity under U50488H-stimulated conditions. GPR88 co-expression (30 ng of cDNA) had a limited influence on β-arr2 recruitment at κOR, contrasting with a significant inhibitory influence on internalization (Kras interaction) at a high dose of GPR88 cDNA, and trafficking (Rab5 interaction) (*Figure 1—figure supplement 2D*). Finally, we evaluated μOR signaling in response to its agonist DAMGO (10 μM) when co-expressed with increasing amounts of GPR88 (*Figure 1—figure supplement 2E*). GPR88 expression had a significant dose-dependent inhibitory effect on DAMGO-activated Gαi/o pathway and β−arr2 recruitment of μOR (see also β−arr1, *Figure 1—figure supplement 2E*), μOR expression remaining stable (*Figure 1—figure supplement 2E*). Focusing on β−arr2/μOR interactions, we observed again that β−arr2 had significantly

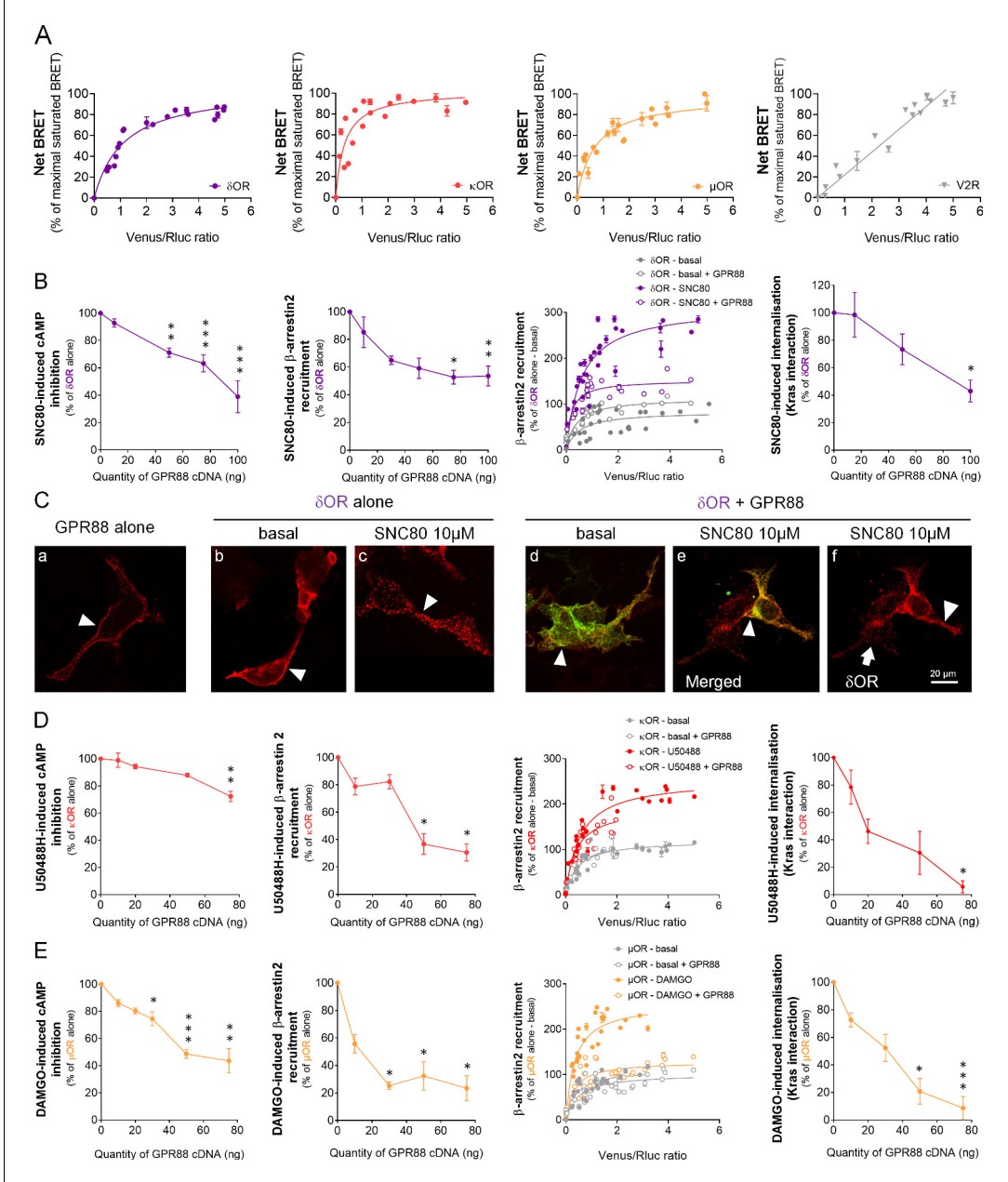

**Figure 1.** GPR88 comes in close physical proximity to opioid receptors and inhibits their signaling and trafficking in vitro. (**A**) BRET1 saturation experiments were performed in transfected HEK293FT cells using constant quantity of GPR88-Rluc8 with increasing amounts of Venus-tagged opioid receptors δOR, κOR and μOR or V2R. Saturated BRET signals indicate close physical proximity (within 10 nm) to the target GPCR, thus possible hetero- or homo-oligomers. Such saturation is not observed with the V2R (right panel), D1 or CXCR4 receptors (see **Figure 4**), showing that close proximity is not detected for all GPCRs. See reverse constructions for δOR, κOR in **Figure 1—figure supplement 1** (**B**) Co-expressing GPR88 with δOR (wild-type or Rluc8-tagged, 30 ng of cDNA) blunts (left to right panels): SNC80 (δOR agonist, 10 μM)-induced inhibition of cAMP production (cAMP sensor: CAMYEL) ($H_{4,39}$=28.7, p=0.0000) and Ypet-β-arrestin 2 (β-arr2) recruitment ($H_{5,28}$=18.1, p=0.0028), Ypet-β-arr2 recruitment at δOR under stimulated but not basal conditions and, finally, SNC80-induced δOR internalization (internalization sensor: Kras-Venus) ($H_{3,12}$=7,8, p=0.0492) in HEK293FT cells. (**C**) Confocal microscopy images show (arrow head) that (a) HA-GPR88 (red) is localized at cell surface (non-permeabilized cells), (b) under basal conditions, HA-δOR (red) is localized at cell surface, (c) SNC80-stimulation induces HA-δOR internalization, (d) under basal conditions, HA-δOR (red) and GPR88-Venus (green) co-localize at cell surface, (e) under SNC80 stimulation, GPR88 inhibits δOR internalization, (f) which results in different patterns of HA-δOR (red) distribution in GPR88-Venus (green) expressing versus non-expressing cells (arrow head versus arrow, respectively). (**D**) Co-expressing GPR88 with κOR (wild-type or Rluc8-tagged, 30 ng of cDNA) decreases (left to right panels): U50488H (κOR agonist, 10 μM)-induced inhibition of cAMP production (CAMYEL) ($H_{4,25}$=17.9, p=0.0013) and Ypet-β-arr2 recruitment ($H_{4,19}$=15.9, p=0.0031), Ypet-β-arr2 recruitment at κOR under stimulated but not basal conditions and U50488H-induced κOR internalization (Kras-Venus) ($H_{4,18}$=13.6, p=0.0087) in HEK293FT cells. (**E**) Co-expressing GPR88 with μOR (wild-

*Figure 1 continued on next page*

*Figure 1 continued*

type or Rluc8-tagged, 30 ng of cDNA) inhibits (left to right panels): DAMGO (µOR agonist, 10 µM)-induced inhibition of cAMP production (CAMYEL) ($H_{5,34}$=27.7, p=0.000) and Ypet-β-arr2 recruitment ($H_{4,23}$=18.2, p=0.0011), Ypet-β-arr2 recruitment at µOR under stimulated but not basal conditions and DAMGO-induced µOR internalization (Kras-Venus) ($H_{4,23}$=19.7, p=0.0006) in HEK293FT cells. Inhibition of cAMP production was determined in presence of 250 µM IBMX and 5 µM forskolin. Data are presented as mean ± SEM of n = 3–9 independent experiments (performed in triplicates). BRET1 values are presented as net BRET (normalized as the percentage of maximal BRET values) or induced BRET (normalized as the percentage of maximal BRET values in absence of GPR88) by Venus/Rluc8 BRET ratio. Asterisks: Kruskal-Wallis ANOVA, multiple comparison of mean ranks, *p<0.05, **p<0.01, ***p<0.001. Confocal imaging: representative pictures among n = 10 pictures. Receptor cell surface expression and additional data regarding OR trafficking in presence of GPR88 are displayed in *Figure 1—figure supplement 2*.

The online version of this article includes the following figure supplement(s) for figure 1:

**Figure supplement 1.** BRET saturation assays using Venus-tagged GPR88 and Rluc8-tagged target receptors.
**Figure supplement 2.** GPR88 inhibits opioid receptor signaling and trafficking in vitro.

more affinity for µOR under stimulated versus basal conditions. Expressing GPR88 reduced the affinity of β−arr2 for DAMGO-stimulated µOR to that measured under basal conditions. In line with these observations, DAMGO-induced internalization (loss of interaction with Kras) and trafficking (interaction with Rab5) of µOR were markedly impacted by the expression of increasing amounts of GPR88. These results indicate that GPR88 can bias the signaling of all three opioid receptors, with µOR appearing to be the most severely impacted.

To further characterize the influence of GPR88 on µOR signaling, we tested the effects of GPR88 co-expression on µOR-induced phosphorylation of ERK in HEK293FT cells. We verified that DAMGO-induced µOR stimulation produced an increase in the phospho-ERK/total ERK total (pERK/tERK) ratio that was suppressed in the presence of GPR88 (*Figure 2A*). Addition of pertussis toxin (PTX) markedly reduced the early phase of DAMGO-induced phosphorylation of ERK, showing its dependence on G protein recruitment ($G_{i/o}$). PTX, however, failed to suppress a later phase of DAMGO-induced response, while GPR88 expression did (*Figure 2B*) (see gels in *Figure 2—figure supplement 1*). This later phase possibly corresponded to incomplete blockade of G-protein recruitment by PTX and/or to G protein-independent phosphorylation of ERK. Thus co-expressing GPR88 with µOR represses phosphorylation of the ERK complex through $G_{i/o}$ protein, further demonstrating the inhibitory influence of GPR88 on G-protein dependent µOR signaling.

We then evaluated how the inhibitory effects of GPR88 on µOR signaling were influenced by the level of expression of the target receptor. To this aim, we first evaluated DAMGO-induced cAMP inhibition by increasing amounts of µOR in presence of a fixed amount of GPR88 (0, 30 or 50 ng of cDNA). We showed that the inhibitory effect of GPR88 on µOR signaling weakens when µOR amounts increase, with a complete restoration for high doses of µOR (*Figure 2C*). Raising the expression of GPR88 shifted this curve to the right, without modifying the maximal effect (full restoration). In contrast, increasing the amount of µOR was not able to completely overcome the inhibitory effect of GPR88 (30 ng of cDNA) on β-arr2 recruitment at µOR; notably, no further recruitment was observed over 50 ng of µOR cDNA transfected. Expressing more GPR88 led to near complete suppression of µOR-dependent β-arrestin recruitment (*Figure 2D*). In conclusion, GPR88 inhibitory action on G-protein dependent µOR signaling was more sensitive to µOR level of expression than its blunting effects on β-arrestin recruitment, suggesting different mechanisms of interaction.

We finally assessed whether GPR88 activation can modulate its ability to interfere with µOR signaling. GPR88 and µOR sharing inhibitory effects on cAMP production (recruitment of a $G_{i/o}$ protein), this made impossible to directly disentangle µOR-dependent signaling from GPR88-dependent activity. However, when subtracting Compound 19-induced cAMP inhibition to the signal measured in presence of both DAMGO and Compound 19, we observed a tendency for an exacerbation of GPR88 blunting effects on µOR-dependent signaling. We then tested the effects of the agonist Compound 19 on GPR88-mediated inhibition of β-arrestin recruitment at µOR in presence of DAMGO. GPR88 similarly repressed DAMGO-induced β-arrestin recruitment by µOR in presence or absence of Compound 19 (*Figure 2—figure supplement 2*).

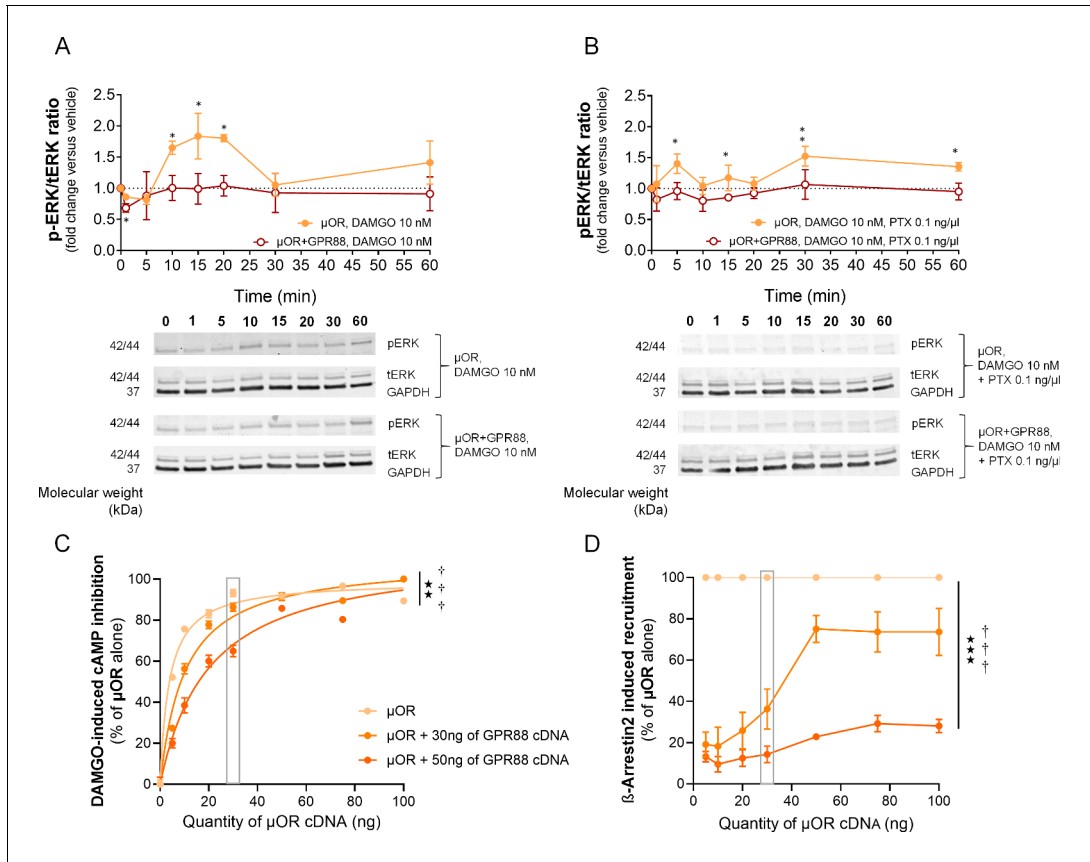

**Figure 2.** GPR88 dampens µOR-mediated signaling in vitro: ERK phosphorylation and effects of µOR expression levels. (**A**) Upper panel: in HEK293FT cells, DAMGO stimulated µOR (0.5 µg cDNA) activity, resulting in an increase in the phospho-ERK/ERK total (pERK/tERK) ratio peaking 15 min after DAMGO stimulation; this response was suppressed when GPR88 (1 µg cDNA) was co-expressed with µOR. Lower panel: representative western blotting images. (**B**) Upper panel: addition of pertussis toxin (PTX, 0.1 ng/µl, overnight) blocked the early phase of DAMGO-induced response, demonstrating its dependence on $G_{i/o}$ protein recruitment, but failed to inhibit a later component of ERK phosphorylation. GPR88 co-expression completely blocked DAMGO-induced phosphorylation of ERK. Lower panel: representative western blotting images. Levels of phosphorylated-ERK (pERK) and total-ERK (tERK) were normalized to the loading control protein glyceraldehyde 3-phosphate dehydrogenase (GAPDH). Data are presented as mean ± SEM of n = 5 independent experiments. Kruskal-Wallis ANOVA, multiple comparison of mean ranks *p<0.05, **p<0.01 (see gels in *Figure 2—figure supplement 1* and statistics in *Supplementary file 1*). (**C**) Increasing the amount of µOR (wild-type, ng of cDNA) expressed in HEK293FT cells allowed to overcome the inhibitory effects of GPR88 (wild-type) on DAMGO (µOR agonist, 10 µM)-induced inhibition of cAMP production (CAMYEL), and a complete rescue of signaling was observed for the highest doses of µOR cDNA transfected (GPR88 effect: $F_{2,21}=86.7$, p=0.0018; µOR effect: $F_{7,147}=90.9$, p=0.0000; GPR88 x µOR interaction: $F_{7,147}=3.1$, p=0.00030). Although increasing the amount of GPR88 (30 to 50 ng of cDNA) shifted the µOR dose response to the right, it was not able to prevent full restoration of µOR signaling at high doses of µOR. (**D**) Increasing the amount of µOR (wild-type) expressed in HEK293FT cells allowed only a partial overcoming of the inhibitory effects of GPR88 (wild-type) on DAMGO (µOR agonist, 10 µM)-induced Ypet-β-arr2 recruitment; no further recruitment was detected for doses of µOR cDNA over 50 ng (GPR88 effect: $F_{2,6}=86.7$, p=0.000037; µOR effect: $F_{6,36}=32.3$, p=0.0000; GPR88 x µOR interaction: $F_{6,36}=15.7$, p=0.0000). Increasing the amount of GPR88 (from 30 to 50 ng of cDNA transfected) nearly suppressed β-arr2 recruitment at µOR. Data for 15 ng of µOR cDNA transfected are framed in gray, to allow comparison with *Figure 1E*. Data are presented as mean ± SEM of n = 3–12 independent experiments (performed in triplicates). BRET1 values are presented as induced BRET (normalized as the percentage of maximal BRET values in absence of GPR88) by Venus/Rluc8 BRET ratio. ANOVA (repeated measure), stars: GPR88 effect, daggers: GPR88xµOR interaction; one symbol: p<0.05, two symbols: p<0.01, three symbols: p<0.001. Effects of GPR88 activation by synthetic agonist Compound 19 on its inhibitory action at µOR signaling in vitro are presented in *Figure 2—figure supplement 2*. The online version of this article includes the following figure supplement(s) for figure 2:

**Figure supplement 1.** Gels from western blot experiments; ERK phosphorylation assay in HEK293FT cells.
**Figure supplement 2.** Effects of Compound 19 on GPR88 signaling pathways and GPR88-mediated blunting of G-protein dependent signaling and β-arrestin recruitment by µOR.

## Deletion of *Gpr88* gene in mice modifies µOR-mediated responses in a behavior-specific manner

Consistent with the above in vitro data, we previously observed an increase in µOR-dependent [$^{35}$S]-GTPγS binding in striatal membranes from *Gpr88$^{-/-}$* mice (*Meirsman et al., 2016a*), suggesting that G protein coupling by µOR was facilitated in the absence of GPR88 expression. We thus hypothesized that *Gpr88* deletion should result in exacerbated morphine-induced µOR-dependent responses in mice. To test this proposition, we assessed several behavioral responses to morphine challenge in *Gpr88$^{-/-}$* mice. The aim of this behavioral screening was to compare morphine-induced behavioral responses depending on the brain regions involved (where GPR88 and µOR receptors are likely or not to co-localize under physiological conditions), the signaling pathways tackled (G protein or β-arrestin-dependent pathways) and GPCR populations engaged (beyond µOR).

First, we measured morphine-induced locomotor activation and its sensitization, two behavioral outcomes tightly depending on striatal function, in *Gpr88* null mice and their wild-type controls after they received daily injections of morphine (IP, 5 mg/kg). The locomotor activity of mutant and control mice was similar under basal (habituation) and vehicle-treated conditions. In both mouse lines, morphine injections induced a significant rise in locomotor activity that progressively sensitized upon repeated administration (*Figure 3A*, panel a). Locomotor activation and sensitization, however, were markedly exacerbated in *Gpr88$^{-/-}$* compared to *Gpr88$^{+/+}$* mice (panel b). These results thus indicate that *Gpr88* deletion facilitates morphine-induced locomotion and sensitization.

Next, we administered morphine (IP, 30 mg/kg) daily for 6 days to a cohort of *Gpr88$^{+/+}$* and *Gpr88$^{-/-}$* mice before triggering pharmacological withdrawal using naloxone (SC, 1 mg/kg). Withdrawal syndrome involves notably activation in the periacqueductal gray, amygdala and nucleus accumbens. We observed that morphine induced-weight loss (measured daily during morphine exposure) was more pronounced in knockout versus control mice (*Figure 3B*, panel a). Following µOR blockade, mutant mice displayed significantly more sniffing episodes (panel b) and exhibited vegetative signs of withdrawal (piloerection, ptosis and teeth chattering) quicker than wild-type controls (panel c), despite comparable global withdrawal scores (panel d) (see more signs in *Figure 3—figure supplement 1*). These data point to exacerbated effects of morphine on withdrawal symptoms, with late signs of withdrawal appearing sooner in *Gpr88* null mice.

Then, we submitted *Gpr88$^{-/-}$* mice and their controls to a conditioned place preference (CPP) paradigm, in which vehicle or morphine injections (SC, 10 mg/kg) were paired with a compartment of the CPP apparatus. This conditioned behavior relies essentially on µOR activation in the ventral tegmental area. Mutant and wild-type animals acquired similarly preference conditioning to morphine (*Figure 3C*, panel a). We then exposed the mice to an extinction protocol, during which exploration of the CPP apparatus was no longer paired with vehicle or morphine injections. Under these conditions, *Gpr88$^{-/-}$* mice extinguished place preference to morphine quicker than their *Gpr88$^{+/+}$* counterparts (panel b). In a last trial, we re-administered morphine (or vehicle) to reinstate place preference: both mouse lines similarly reinstated preference for the reinforced compartment (panel c). Thus, *Gpr88* deletion had no effect on morphine-induced CPP and reinstatement, suggesting preserved morphine reward in knockout animals, but reduced the lapse of time they needed to extinguish this conditioning.

In a next step, we assessed the nociceptive thresholds of *Gpr88* mutants and controls in the tail immersion and hot-plate tests under vehicle or morphine (IP, 5 mg/kg) challenge. The first test involves spinal responses whereas thermic nociception depends on central amygdala function. Responses to nociceptive stimuli did not differ between *Gpr88$^{-/-}$* and *Gpr88$^{+/+}$* mice under vehicle conditions. In the tail immersion test, however, while we detected significant analgesic effects of morphine across temperatures in *Gpr88$^{+/+}$* animals, this response was markedly blunted in *Gpr88$^{-/-}$* mice (*Figure 3D*, panel a). In contrast, in the hot-plate test, jumping latency under morphine challenge, but not licking latency, was longer in mutant compared to wild-type animals, indicating that supraspinal morphine-induced analgesia was facilitated in *Gpr88* null mice (panel b). Thus, the behavioral consequences of *Gpr88* deletion on morphine-induced responses differed depending on the behavior assessed, pointing to brain substrates, signaling pathways and additional GPCRs involved as key modulators of these responses.

Having shown above that mice lacking *Gpr88* display a drastic increase in their locomotor sensitization to repeated morphine administration, and that this effect was highly significant since the

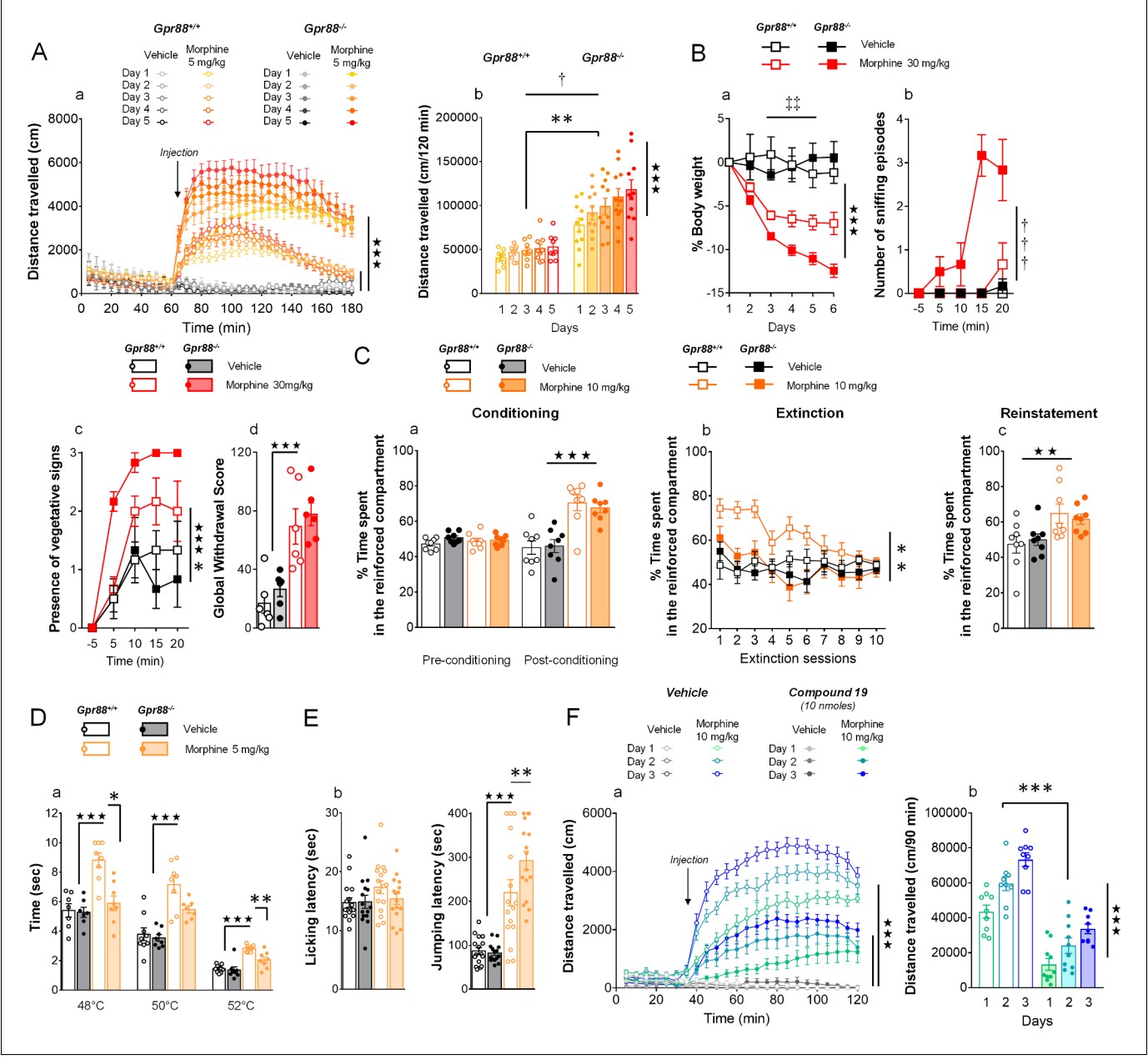

**Figure 3.** *Gpr88* null mice display modified mu-opioid mediated behavioral responses. (**A**) In a locomotor sensitization paradigm (n = 5 to 11 mice per treatment and genotype) morphine induced an increase in locomotor activity that sensitized upon repeated administration in *Gpr88⁺/⁺* and *Gpr88⁻/⁻* mice (panel a); morphine-induced locomotion and sensitization, however, were significantly greater in mutant mice (panel b) (Genotype effect: $F_{1,27}$=13.5, p=0.0010; Treatment: $F_{1,27}$=99.9, p=0.0000; Genotype x Treatment interaction: $F_{1,27}$=14.6, p=0.0007; Day: $F_{4,108}$=7.9, p=0.0000; Day x Treatment: $F_{4,108}$=10.2, p=0.0000; Day x Genotype x Treatment = 2.9, p=0.0253); solid stars: treatment effect (two-way ANOVA with one repeated measure – day), asterisks: genotype effect, dagger: Day x Genotype x Treatment interaction. (**B**) Upon exposure to escalating doses of morphine (n = 6 per treatment and genotype), Gpr88⁻/⁻ lost more body weight than controls (panel a; Treatment: $F_{1,20}$=43.7, p=0.0000; Day: $F_{4,80}$=13.6, p=0.0000; Day x Genotype: $F_{4,80}$=5.1, p=0.001; Day x Treatment: $F_{4,80}$=9.9, p=0.0000; body weight was measured daily upon morphine treatment); when withdrawal was triggered by acute naloxone administration (1 mg/kg), mutant animals displayed sniffing episodes (panel b; Genotype/Treatment: $F_{1,20}$=25.5, p=0.0000; Genotype x Treatment: $F_{1,20}$=23.1, p=0.0001; Time: $F_{3,60}$=10.5, p=0.0000; Time x Genotype/Treatment: $F_{3,60}$=9.3, p=0.0000; Time x Genotype x Treatment: $F_{3,60}$=8.3, p=0.0001) and vegetative signs of withdrawal (panel c; Genotype: $F_{1,20}$=5.3, p=0.0324; Treatment: $F_{1,20}$=5.3, p=0.0324; Time: $F_{3,60}$=14.4, p=0.0000) quicker than their *Gpr88⁺/⁺* counterparts, despite similar final withdrawal scores (panel d; Treatment: $F_{1,20}$=38.9, p=0.0000); solid stars: treatment effect (two-way ANOVA with one repeated measure – day/5 min time bin), double daggers: Time x Genotype interaction, daggers: Time x Genotype x Treatment interaction, asterisk: genotype effect. More withdrawal signs are displayed in *Figure 3—figure supplement 1*. (**C**) In a CPP paradigm (n = 8 per treatment and genotype), *Gpr88⁻/⁻* mice acquired preference for a compartment associated to morphine administration (10

*Figure 3 continued on next page*

Figure 3 continued

mg/Kg) similarly as $Gpr88^{+/+}$ animals (panel a; Treatment: $F_{1,28}=45.8$, p=0.0000; Conditioning: $F_{1,28}=15.7$, p=0.0005; Conditioning x Treatment: $F_{1,28}=31.1$, p=0.0000); they extinguished this conditioning quicker than wild-type counterparts (panel b; Genotype: $F_{1,28}=10.3$, p=0034; Treatment: $F_{1,28}=6.5$, p=0166; Genotype x Treatment: $F_{1,28}=5.0$, p=0329; Session: $F_{9,252}=5.2$, p=0.0000; Session x Treatment: $F_{9,252}=3.7$, p=0.0002) and finally reinstated morphine place preference at comparable levels as the latter (panel c; Treatment: $F_{1,28}=9.2$, p=0.0052); solid stars: Treatment effect (two-way ANOVA with one repeated measure – day/5 min time bins); asterisks: Genotype effect. (D) In the tail flick test (n = 7–9 per treatment and genotype), Gpr88$^{-/-}$ mice were significantly less sensitive to morphine analgesia at 48°C and 52°C (Genotype: $F_{1,83}=22.0$, p=0.0000; Treatment: $F_{1,83}=84.9$, p=0.0000; Temperature: $F_{2,83}=162.5$, p=0.0000; Genotype x Treatment: $F_{1,83}=15.9$, p=0.0001; Treatment x Temperature: $F_{2,83}=5.4$, p=0.0065); (E) in the hot plate test (n = 15–16 per treatment and genotype), morphine-induced analgesia was detected by increased jumping latency in treated animals (right panel); this effect was increased in Gpr88 null mice versus controls (Treatment: $F_{1,59}=79.7$, p=0.0000; Genotype x Treatment: $F_{1,59}=4.0$, p=0.0490); solid stars: Treatment effect (two-way ANOVA), asterisks: Genotype x Treatment interaction (Newman Keules post-hoc test). Increased morphine-induced locomotor sensitization in Gpr88 null mice was not associated with modified pERK/tERK ratio in three brain regions (**Figure 3—figure supplements 2 and 3**). (F) We administered the GPR88 agonist Compound 19 (icv, 10 nmoles) to mice exposed to a morphine-induced locomotor sensitization paradigm (n = 8 to 9 mice per treatment condition). Exposure to morphine (IP, 10 mg/kg) induced an increase in locomotor activity that sensitized upon repeated administration in both vehicle and Compound 19-treated groups (panel a); pharmacological activation of GPR88 drastically reduced morphine-induced locomotor activity but left the amplitude of sensitization unchanged (Morphine effect: $F_{1,30}=305.1$, p=0.0000; Compound 19: $F_{1,30}=55.3$, p=0.0000; Day: $F_{2,60}=33.1$, p=0.0000; Day x Morphine: $F_{2,60}=31.6$, p=0.0000; Day x Morphine x Compound 19: $F_{2,60}=2.1$, NS), solid stars: treatment effect (two-way ANOVA with one repeated measure – day), asterisks: genotype effect. Data are presented as mean ± SEM. One symbol: p<0.05, two symbols: p<0.01, three symbols: p<0.001.

The online version of this article includes the following figure supplement(s) for figure 3:

**Figure supplement 1.** Additional morphine-induced withdrawal signs in $Gpr88^{-/-}$ versus $Gpr88^{+/+}$ mice.

**Figure supplement 2.** Increased morphine-induced locomotor sensitization in Gpr88 null mice was not associated with modified pERK/tERK ratio in three brain regions.

**Figure supplement 3.** Gels from western blot experiments; ERK phosphorylation assay in brain samples from $Gpr88^{+/+}$ and $Gpr88^{-/-}$ mice.

second administration of morphine, we aimed at evaluating μOR signaling by assessing pERK/tERK ratio in brain regions from $Gpr88^{-/-}$ versus $Gpr88^{+/+}$ mice, in an attempt to capture β-arr2/pERK recruitment, shown to play a crucial role in morphine-induced locomotor sensitization (**Tao et al., 2017**; **Urs et al., 2011**; **Valjent et al., 2005**; **Valjent et al., 2010**). We exposed $Gpr88^{+/+}$ and $Gpr88^{-/-}$ mice to morphine administration (IP, 5 mg/kg) on two consecutive days in the same environment (**Valjent et al., 2010**). We observed on the second day a robust locomotor sensitization in mutant animals that was not observed in wild-type animals (**Figure 3—figure supplement 2A**). 60 min after second morphine injection, however, we failed to detect differences in the pERK/tERK ratio between $Gpr88^{-/-}$ and $Gpr88^{+/+}$ animals in striatal regions (caudate putamen and nucleus accumbens), or in the periaqueductal gray (**Figure 3—figure supplement 2B**) (see gels in **Figure 3—figure supplement 3**). Therefore, under these experimental conditions, we could not detect a significant impact of Gpr88 deletion on μOR-mediated β-arr2/pERK ex vivo.

Finally, we addressed the question of potential effects of GPR88 pharmacological activation on its ability to interfere with μOR signaling by evaluating the consequences of Compound 19 administration (icv, 10 nmoles) on morphine (IP, 10 mg/kg)-induced locomotion and sensitization in wild-type mice (**Figure 3F**). Compound 19 markedly inhibited morphine-induced locomotion in these animals along the three days of morphine exposure (**Figure 3F**, panel a). However, the amplitude of morphine-induced locomotor sensitization was similar in mice that received the GPR88 agonist or vehicle (**Figure 3F**, panel b), indicating that Compound 19 had no detectable influence on this process. Thus, pharmacological activation of GPR88 can potentiate some, although not all, inhibitory effects of this orphan receptor on μOR signaling in vivo.

## GPR88 comes in close proximity to multiple GPCRs and interferes with their signaling

Morphine-induced behavioral responses rely on complex molecular mechanisms and the activation of multiple GPCRs beyond μOR. We thus explored whether GPR88 may come in close physical vicinity of other GPCRs, focusing first on GPCRs with striatal expression, such as muscarinic (M1, M4), dopaminergic (D1, D2), adenosine ($A_{2A}$) or orphan receptor GPR12 (**Ho et al., 2018**; **Heiman et al., 2008**), and then extending our interest to GPCRs not known to be expressed in CNS neurons, namely the vasopressin V2R and chemokine CXCR4 receptors. Remarkably, GPR88 displayed saturated BRET signals with all tested striatal GPCRs except the dopamine D1 receptor (**Figure 4A**). In

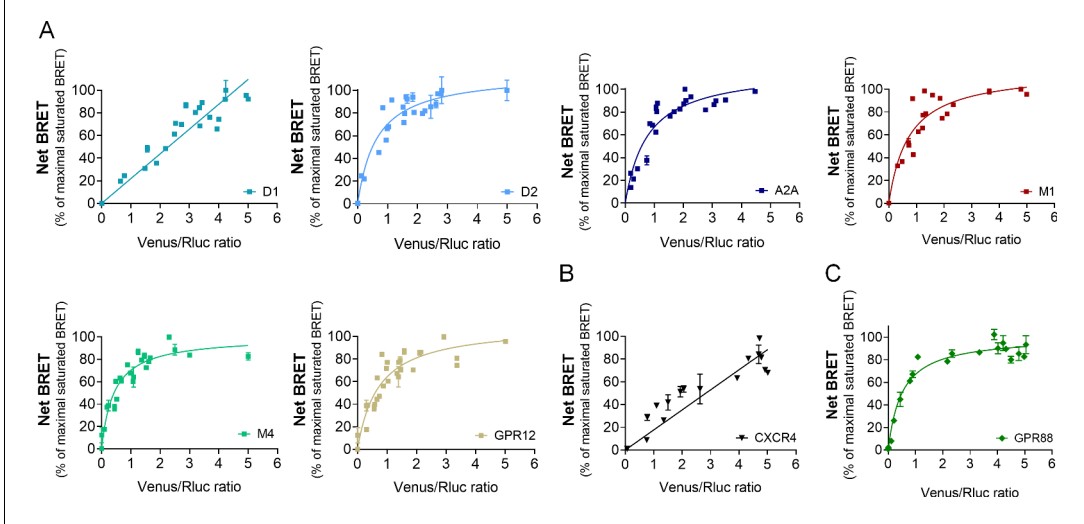

**Figure 4.** GPR88 comes in close proximity to multiple GPCRs. BRET1 saturation experiments were performed in transfected HEK293FT cells using constant quantity of GPR88-Rluc8 with increasing amounts of (**A**) Venus-tagged striatal GPCRs: dopamine D1 and D2, adenosine $A_{2A}$, muscarinic M1 and M4, and orphan receptor GPR12, (**B**) Venus-tagged non neuronal GPCRs: chemokine CXCR4 and V2R (see **Figure 1**). (**C**) Venus-tagged GPR88. Saturated BRET signals indicate close physical proximity (within 10 nm) to the target GPCR. GPR88 comes in close proximity and possibly forms hetero-oligomers with D2, $A_{2A}$, M1, M4, GPR12 and itself, but not D1, V2R and CXCR4 receptors. Values (mean ± SEM) from n = 3–4 independent experiments (performed in triplicates) are presented as net BRET (normalized as the percentage of maximal BRET values) by Venus/Rluc8 BRET ratio.

contrast, BRET signal was unsaturated for interactions with the two GPCRs identified as not expressed in CNS neurons, V2R and CXCR4 receptors (**Figure 4B**). Thus GPR88 appears to possibly form hetero-oligomers with multiple GPCRs in addition to opioid receptors, notably those GPCRs whose expression is enriched in striatal regions, where GPR88 is also the most expressed.

We next explored the functional consequences of GPR88 co-expression on the signaling of striatal and non-striatal GPCRs. We co-expressed GPR88 with these GPCRs and assessed the effects of such co-expression on their ability to activate their G protein and β-arrestin dependent signaling pathways. Interestingly, increasing amounts of GPR88 expression interfered with the G protein mediated signaling of all GPCRs from which it comes close (dopamine D2 and muscarinic M1 and M4) except for the adenosine $A_{2A}$ receptor (**Figure 5A**, panels a,c,d,e,f). In contrast, co-expressing GPR88 with receptors for which we failed to evidence proximity to it, namely dopamine D1, vaso-pressin V2R and chemokine CXCR4 receptors, had no significant effect on their G protein dependent signaling, or even increased this signaling (D1 receptor) (**Figure 5A**, panels b,g,h). Of note, the impact of GPR88 expression on G protein mediated signaling was observed independently from the nature of the G protein coupled by the target receptor, either Gαs (M1, M4), Gαi/o (D2) or Gαq/11 (M4). When now focusing on β-arrestin recruitment, the picture was remarkably homogeneous: expressing increasing amounts of GPR88 dose-dependently dampened β-arr2 recruitment by all GPCRs tested (**Figure 5B**, panels a-g). Moreover, GPR88 expression markedly blunted CXCR4-dependent β-arr2 recruitment under both basal and stimulated conditions (**Figure 5B**, panel h). Taken together, our functional results indicate that GPR88 expression represses the G protein-dependent signaling of multiple GPCRs, more likely when it possibly forms hetero-oligomers with them, whereas it inhibits β-arr2 recruitment for all GPCRs tested.

## GPR88 blunts β-arrestin recruitment by other GPCRs independently from physical proximity

In an attempt to better understand how GPR88 impacts the signaling of other GPCRs, we compared the consequences of co-expressing GPR88 on the activation of the G-protein dependent pathway and the recruitment of β-arrestins by μOR, with which GPR88 potentially forms hetero-oligomers (**Figure 6A**), and CXCR4, for which we could not detect proximity to GPR88 (**Figure 6B**). Moreover,

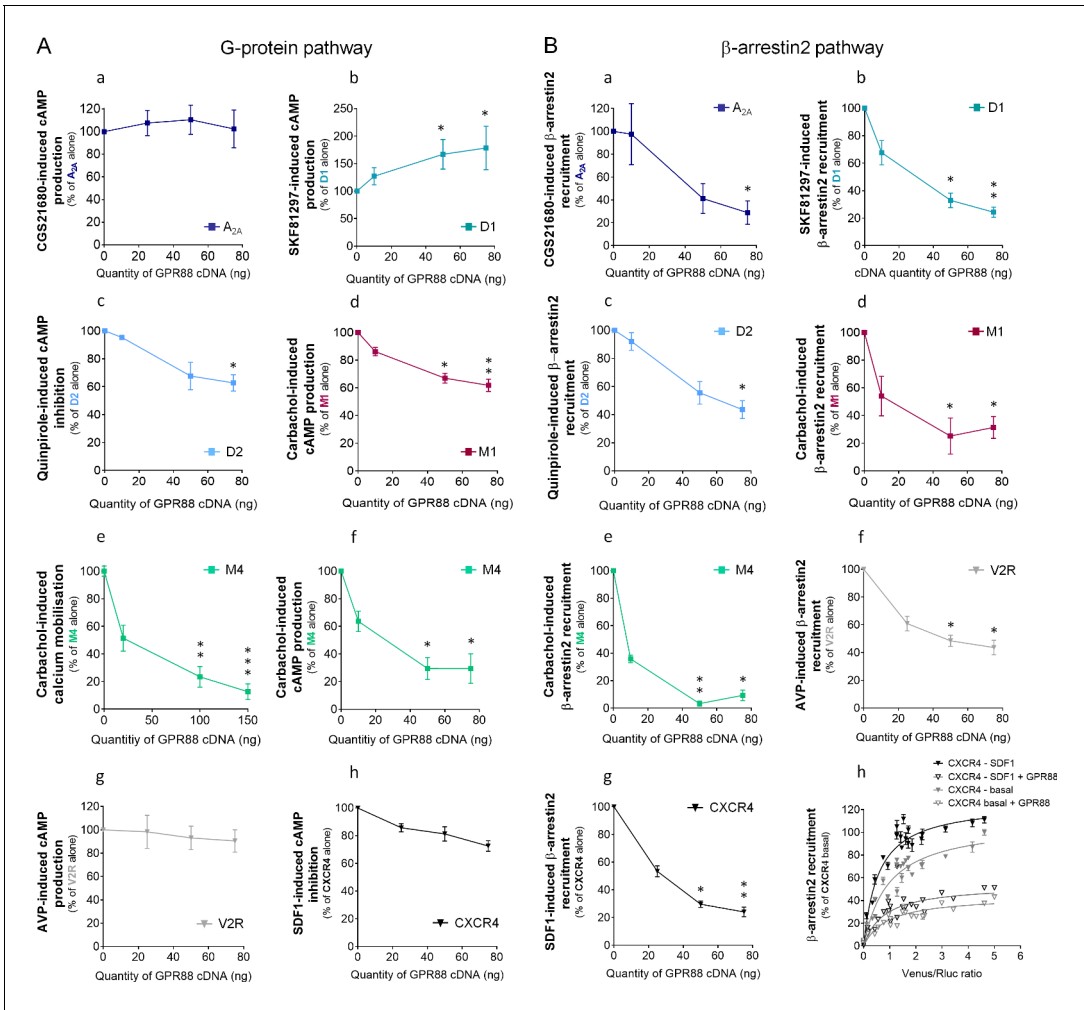

**Figure 5.** GPR88 biases the signaling of multiple GPCRs. We evaluated the consequences of GPR88 co-expression on the signaling of adenosine $A_{2A}$, dopamine D1, dopamine D2, muscarinic M1, muscarinic M4, vasopressin V2R and CXCR4 receptors in HEK293FT cells. BRET1 assay was used to assess the activation of the (**A**) G protein dependent pathway (cAMP sensor: CAMYEL; calcium sensor: aequorin-GFP) or (**B**) the recruitment of Ypet β-arrestin 2 (β-arr2) by Venus-tagged receptors in presence of increasing amounts of GPR88 cDNA transfected. (**A**) GPR88 co-expression blunted the G protein mediated signaling of all GPCRs from which it comes close (panels c-f; dopamine D2: $H_{3,12}$=9.6, p=0.0223; muscarinic M1: $H_{3,16}$=13.2, p=0.0041; M4 calcium: $H_{3,35}$=24.0, p=0.0000; M4 cAMP: $H_{3,12}$=12.9, p=0.0048), except the adenosine $A_{2A}$ receptor (panel a; $H_{3,12}$=1.2, p=0.7602). GPR88 had no significant impact on, or even facilitated, G protein dependent signaling of receptors for which we failed to evidence proximity to it, namely D1 (panel b, increased activity in presence of GPR88; $H_{3,20}$=9.7; p=0.0211), V2R and CXCR4 receptors (panels g,h; V2R: $H_{3,16}$=1.2, p=0.7602; CXCR4: $H_{3,12}$=7.6, p=0.0546). (**B**) In contrast, GPR88 co-expression compromised the ability of all GPCRs tested to recruit Ypet-β-arr2 when activated by their agonist, independently from previously evidenced close proximity to GPR88 (panels a-g; $A_{2A}$: $H_{3,12}$=9.7, p=0.0211; D1: $H_{3,12}$=9.6, p=0.223; D2: $H_{3,16}$=13.2, p=0.0080; M1: $H_{3,15}$=10.4, p=0.0151; M4: $H_{3,16}$=13.2, p=0.0041; V2R: $H_{3,16}$=10.9, p=0.0123; CXCR4: $H_{3,16}$=13.2, p=0.0041). Focusing on CXCR4, co-expressing GPR88 diminished the probability of Ypet-β-arr2 recruitment at this receptor under both stimulated and basal conditions (panel h). Specific agonists used to stimulate GPCRs were: CGS21680 ($A_{2A}$, 10 μM), SKF81297 (D1, 10 μM), quinpirole (D2, 10 μM), carbachol (M1 and M4, 10 μM), AVP (V2R, 10 μM) and SDF1 (CXCR4, 125 μM). Data are presented as mean ± SEM of n = 3–4 independent experiments (performed in triplicates). BRET1 values are presented as induced BRET (normalized as the percentage of maximal BRET values in absence of GPR88) normalized by Venus/Rluc8 BRET ratio. Asterisks: Kruskal-Wallis ANOVA, multiple comparison of mean ranks, *p<0.05, **p<0.01.

the effects of GPR88 co-expression were compared to those of co-expressing the alternative GPCR (CXCR4 for μOR, and reciprocally) or κOR, as examples of GPCRs forming or not heterodimers with the target receptor. Indeed, we verified that μOR and κOR closely interact (*Figure 6C*, upper panel), as previously reported (*Fujita et al., 2014*), and evidenced that CXCR4 likely forms hetero-oligomers with κOR but not μOR (*Figure 6C*, middle and right panels, respectively). Consistent with no detected proximity, CXCR4 co-expression had no deleterious effect on DAMGO-induced μOR signaling, and conversely μOR co-expression did not modify SDF1-mediated CXCR4 signaling, on either

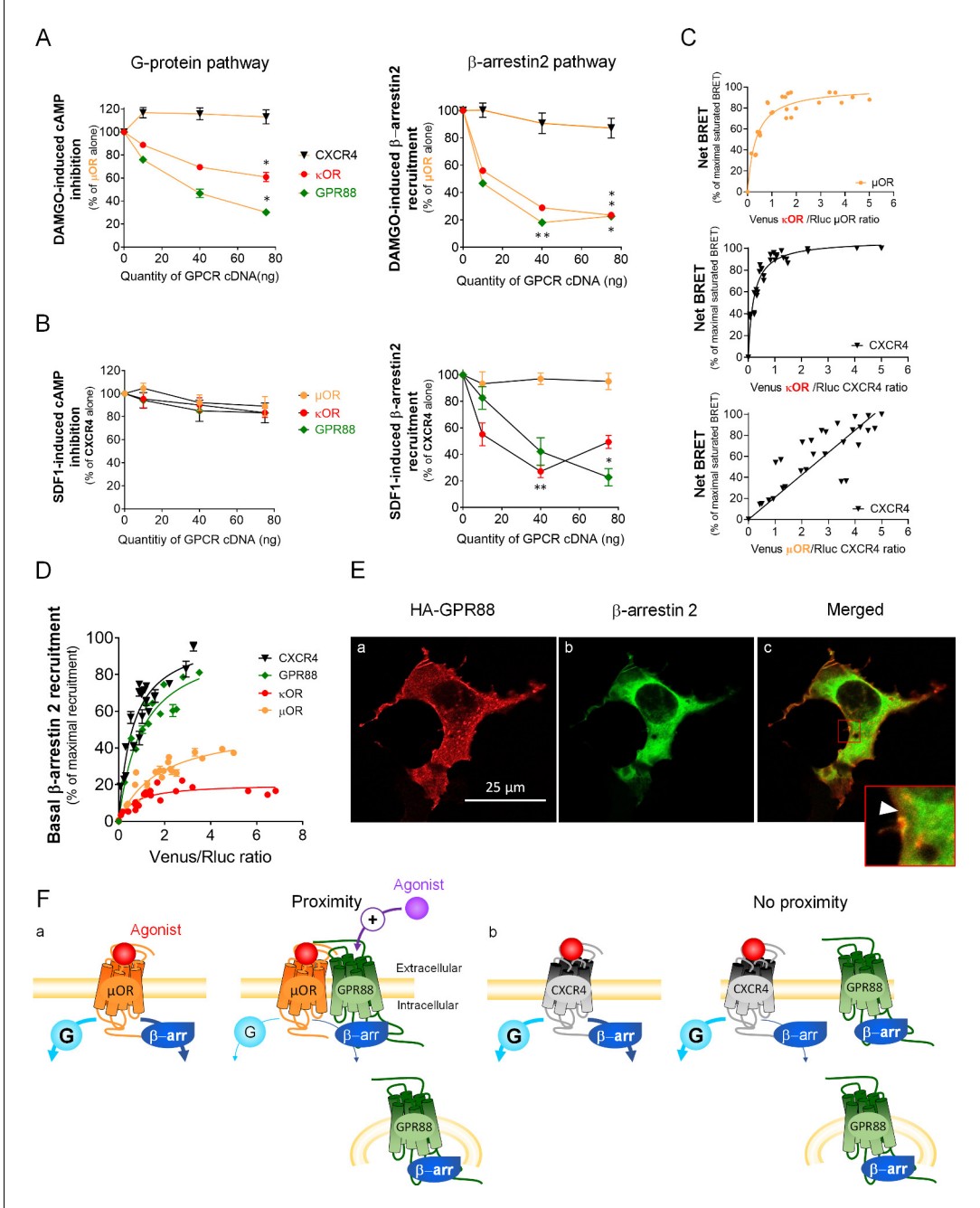

**Figure 6.** GPR88 impedes the recruitment of β-arrestins at other GPCRs independently from physical proximity. (**A**) We evaluated the functional consequences of co-expressing CXCR4, κOR and GPR88 with μOR on DAMGO-induced cAMP production and β-arr2 recruitment by μOR. CXCR4 co-expression had no deleterious impact on G-protein-dependent signaling and the recruitment of β-arrestins (cAMP: $H_{3,16}$=6.6, p=0.0840; β-arr2: $H_{3,16}$=2.8, p=0.427); in contrast, κOR and GPR88 expression dampened the activation of both, with GPR88 inhibiting cAMP production by μOR more efficiently than κOR (cAMP - κOR: $H_{3,12}$=10.1, p=0.0176; GPR88: $H_{3,12}$=10.5, p=0.0145; β-arr2 - κOR: $H_{3,16}$=13.8, p=0.0031; GPR88: $H_{3,16}$=13.5, p=0.0036). (**B**) We then assessed the consequences of co-expressing μOR, κOR and GPR88 with CXCR4 on SDF1-induced cAMP production and β-arr2 recruitment by CXCR4. None of the co-expressed receptors had a significant impact on cAMP production by CXCR4 (μOR: $H_{3,12}$=7.3, p=0.064; κOR: $H_{3,12}$=2.6, p=0.4593; GPR88: $H_{3,12}$=2.9, p=0.4077); as regards β-arr2 recruitment, μOR co-expression had no influence ($H_{3,12}$=0.75, p=0.8604), whereas both κOR ($H_{3,16}$=12.7, p=0.0053) and GPR88 ($H_{3,12}$=9.8, p=0.0203) reduced it, with GPR88 having a more significant influence than κOR at the highest dose transfected. (**C**) BRET1 saturation experiments evidenced close proximity between μOR and κOR (upper panel), CXCR4 and κOR (middle panel) but not CXCR4 and μOR (lower panel). (**D**) Ypet-β-arr2 recruitment was higher at CXCR4 and GPR88 than κOR and μOR under basal conditions. (**E**) Confocal microscopy images show that (**a**) HA-GPR88 is localized at cell surface and also in the cytosol, following a patchy distribution (permeabilized cells), (**b**) β-arr2 is expressed in a diffuse pattern near the cell membrane and almost all the cytoplasm and (**c**) Colocalisation of GPR88 and β-arr2 is

*Figure 6 continued on next page*

*Figure 6 continued*

clearly seen at the level of cytoplasmic patches, with the intensity of β-arr2 expression appearing lower around the densest GPR88-expressing patches (framed, arrow head). (F) Schematic representation of the putative mechanisms of GPR88 inhibitory action at GPCR signaling. Panel a: GPR88 dampens G-protein-mediated signaling of GPCRs to which it comes close (such as μOR), likely by interfering with G-protein coupling, and inhibits the recruitment of β-arrestins by sequestering them in intracellular compartments. Pharmacological activation of GPR88 may potentiate its inhibitory action on G-protein dependent signaling only. Panel b: when no proximity is detected between GPR88 and the target receptor (such as CXCR4), only the effects on β-arrestin recruitment are observed. Data (A–D) are presented as mean ± SEM of n = 3–4 independent experiments (performed in triplicates). BRET1 values are presented as net BRET (normalized as the percentage of maximal BRET values) or induced BRET (normalized as the percentage of maximal BRET values when the target receptor is expressed alone) by Venus/Rluc8 BRET ratio. Asterisks: Kruskal-Wallis ANOVA, multiple comparison of mean ranks, *p<0.05, **p<0.01. Confocal imaging (E): representative pictures among n = 10 pictures. Legend to *Supplementary file 1*.

the G protein- or β-arrestin-dependent pathways (*Figure 6A–B*). In contrast, κOR expression repressed agonist-mediated G protein activation and β-arr2 recruitment by μOR, with which it forms hetero-oligomers (*Figure 6A*). Such expression however had no significant impact on G protein-mediated signaling by CXCR4 but significantly inhibited β-arr2-dependent recruitment by this receptor (*Figure 6B*). Compared to κOR, GPR88 had a more significant inhibitory impact on G protein dependent μOR signaling but similarly blunted μOR-dependent β-arr2 recruitment (*Figure 6A*). Finally, although GPR88 had little influence on SDF1-induced G protein activation by CXCR4 (see also *Figure 5G*), it was able to repress CXCR4-dependent β-arr2 recruitment more than κOR, despite no detectable physical proximity to CXCR4 (*Figures 4B* and *6B*). Together, these results indicate that GPR88 can interfere with the signaling of other GPCRs when coming in close vicinity of them, but also in the absence of detected proximity.

A mechanism through which GPR88 may dampen GPCR recruitment of β-arrestins would be by sequestering them away from the plasma membrane, as shown for vasopressin V2R and neurokinin NK1 receptors upon pharmacological activation (*Klein et al., 2001*; *Schmidlin et al., 2002*). Under these conditions, however, V2R and NK1 display a high affinity for β-arrestins and an intracellular distribution. Here, we evaluated the affinity for β-arrestins of GPR88 compared to μOR, δOR and CXCR4 under basal conditions (no agonist). The probability that β-arr2 was recruited at GPR88 and CXCR4 in the absence of pharmacological stimulation was higher than at the two opioid receptors (*Figure 6D*). We then used immunochemistry to assess the respective intracellular distributions of HA-GPR88 and β-arr2 after membrane permeabilization. We observed, in addition to location at cell membrane, a significant, patchy, intracellular distribution of GPR88 (*Figure 6Ea*), suggesting its presence in vesicular compartments. Diffuse cytosolic distribution of β-arr2 (*Figure 6Eb*) was found colocalized with GPR88-expressing patches (*Figure 6Ec*) and appeared depleted around the brightest ones (framed, arrow head), suggesting sequestration by GPR88. Thus, GPR88 can bias the signaling of other GPCRs independently from oligomerization, possibly by trapping β-arrestins in intracellular compartments.

## Discussion

In the present study, we evidence for the first time physical proximity between GPR88 and the three opioid receptors, and a negative impact of GPR88 expression on opioid-receptor mediated G protein and β-arrestin recruitment in vitro, μOR-dependent signaling appearing the most severely affected. In mice, *Gpr88* deletion produced different effects on morphine-induced (μOR-dependent) responses depending on the behavior assessed, exacerbating morphine-induced locomotor sensitization, withdrawal syndrome and supra-spinal analgesia while blunting morphine effects on spinal nociceptive responses and leaving CPP unchanged. Importantly, we also detected close proximity between GPR88 and multiple other GPCRs with enriched striatal expression but not with GPCRs not expressed in CNS neurons. GPR88 co-expression was able to inhibit G protein dependent signaling of most GPCRs from which it comes close, and only these. In contrast, such co-expression resulted in blunted β-arrestin recruitment by all GPCRs tested.

The first finding from this study is that GPR88 closely interacts with opioid receptors and that its expression impedes opioid signaling in vitro. Saturated BRET signals of GPR88 in presence of δOR, κOR and μOR indicate physical proximity with opioid receptors and suggest potential hetero-oligomerization. Remarkably, although GPR88 displayed similar saturated BRET signals in presence of the

three opioid receptors, the orphan receptor seemed to have a differential influence on their signaling. GPR88 co-expression had only a modest impact on κOR-mediated G protein pathway activation compared to δOR and μOR. Conversely, δOR appeared to be the less affected among opioid receptors on its ability to recruit β-arrestins, as seen from agonist-induced β-arr2 recruitment and receptor internalization. Finally, μOR function seemed to be the most severely altered by GPR88 co-expression, on both G protein and β-arrestin-dependent pathways.

In order to better characterize the effects of GPR88 expression on opioid signaling, we focused on GPR88-μOR functional interactions. We first confirmed the inhibitory effect of GPR88 expression on G-protein dependent signaling of μOR, by evidencing a suppression of $G_{i/o}$ protein-dependent μOR-induced phosphorylation of ERK in presence of GPR88. We then evaluated how μOR expression levels impact the effects of GPR88 co-expression on G-protein dependent signaling and β-arrestin recruitment by μOR. We observed a weakening of the inhibitory action of GPR88 on μOR-mediated G-protein dependent signaling when μOR expression increased over GPR88 expression. Thus, direct μOR-GPR88 interactions seem necessary for GPR88 to exert its inhibitory effects; moreover, the regular pace of the rightwards shift of μOR signaling dose response when GPR88 amounts increase suggests a 1:1 (or n/n) stoichiometry in this interaction. In contrast, increasing μOR amounts failed to fully restore μOR-mediated β-arrestin recruitment in presence of GPR88, suggesting that a pool of β-arrestins remained out of reach of μOR, regardless μOR expression levels. Therefore, GPR88 effects on μOR-mediated G-protein dependent signaling and β-arrestin recruitment likely involved different mechanisms of interaction. Interestingly, differential relative inhibitory effects of GPR88 on the signaling of the different opioid receptors (and other GPCR partners) possibly reflected differential sensitivity to variations in the expression of the partner receptor, G-protein signaling being more sensitive than β-arrestin recruitment to such variations. Finally, we tested whether agonist-induced activation of GPR88 influences its effects on μOR signaling in vitro. Activation of GPR88 by Compound 19 tended to exacerbate the inhibitory action of GPR88 on μOR-mediated G-protein dependent signaling, but had no detectable effect on β-arrestin recruitment by μOR. These results suggest that GPR88 is more efficient in blunting GPCR partner's G-protein dependent signaling, but not β-arrestin recruitment, if in an active-like conformation.

The physiological consequences of GPR88 expression on opioid signaling are not straightforward to predict from in vitro experiments, as inhibition of G protein signaling argues for a reduction of (δOR and μOR) opioid signaling, whereas dampened β-arrestin recruitment suggests a retention of opioid receptors at the cell surface and maintained signaling but loss of β-arrestin-mediated functional effects. We took advantage of analyzing the phenotype of *Gpr88* null mice to get insight into interactions between GPR88 and opioid signaling in vivo. Our previous findings of increased δOR-mediated G protein signaling in striatal membranes of *Gpr88*[-/-] mice together with a phenotypic profile opposing that of mice lacking δOR (*Le Merrer et al., 2013*) and partially normalized by δOR antagonist administration (*Meirsman et al., 2016a*) plead for an excessive δOR activity in these animals, consistent with a main inhibitory impact of GPR88 on the G protein dependent signaling of this receptor. In the present study, we evaluated the in vivo consequences of *Gpr88* deletion on μOR signaling by challenging knockout animals with morphine across several experimental paradigms. Morphine-induced responses in *Gpr88*[-/-] knockouts were differentially modified depending on the behavior assessed, likely due to differences in the brain regions, signaling pathways and GPCR populations engaged to mediate these responses. The prominent factor influencing the effects of *Gpr88* deletion, however, was likely the brain substrates underlying behavioral responses, and whether GPR88 is expressed in these structures under physiological conditions. Indeed, *Gpr88* deletion exacerbated morphine-induced responses that involved μOR activation in brain regions where *Gpr88* expression is usually enriched (*Massart et al., 2016*; *Ehrlich et al., 2018a*; *Becker et al., 2008*), namely locomotor activity and sensitization, depending on μOR signaling in the striatum (*Tao et al., 2017*; *Charbogne et al., 2017*), withdrawal symptoms involving μOR activity in the nucleus accumbens (*Williams et al., 2001*; *Le Merrer et al., 2009*) and supra-spinal analgesia measured in the hot-plate test, modulated notably by μOR activity in the central amygdala (*Pavlovic et al., 1996*; *Pavlovic and Bodnar, 1998*). In contrast, morphine-induced CPP and reinstatement were not modified in *Gpr88*[-/-] mice. Interestingly, morphine-induced CPP involves primarily μOR signaling in the ventral tegmental area (*Charbogne et al., 2017*; *Le Merrer et al., 2009*) where GPR88 is scarcely expressed (*Ehrlich et al., 2018a*; *Ehrlich et al., 2018b*). This could account for unchanged morphine CPP in the absence of *Gpr88* expression. Of note, *Gpr88* null animals

extinguished CPP faster than controls, in agreement with previously evidenced facilitation of hippo-campus-dependent learning processes (*Meirsman et al., 2016a*). A second factor that plausibly influenced morphine-induced responses in *Gpr88*[-/-] mice was the signaling pathway involved in μOR-mediated effects. Indeed, locomotor sensitization, markedly exacerbated in knockout mice, was evidenced to depend tightly on β-arrestin recruitment not only by μOR, but also by the D1 dopamine receptor (*Tao et al., 2017*; *Urs et al., 2011*; *Borgkvist et al., 2008*; *Becker et al., 2001*). As regards these behavioral responses, the phenotypic profile of *Gpr88* mutant animals is thus consistent with a relief of GPR88 break on β-arrestin recruitment by μOR, and D1. Now focusing on nociception, morphine-induced analgesia was significantly reduced in the tail immersion test, which tackles spinal function (*Ramabadran et al., 1989*). *Gpr88* expression in spinal cord is discrete (*Massart et al., 2016*; *Becker et al., 2008*). Its deletion, however, may have allowed massive μOR-mediated β-arrestin recruitment under morphine stimulation, which was shown to be detrimental to spinal morphine analgesia (*Yang et al., 2011*). Finally, behavioral responses to morphine in *Gpr88*[-/-] mice may not be the consequence of GPR88 removal on μOR function only but also on the signaling of other GPCRs involved in the expression of these responses. Notably, morphine-induced locomotor sensitization involves striatal dopamine and cholinergic receptors (*Tao et al., 2017*; *Valjent et al., 2005*; *Ruan et al., 2019*), and the effects of GPR88 deletion on their pharmacology likely contributed to the exacerbation of this phenomenon in mutant animals. The involvement of multiple GPCR players in morphine-induced responses may have participated to our failure in detecting the expected increase in pERK/tERK ratio in *Gpr88* knockout mice sensitized to morphine; modified kinetics of ERK phosphorylation in vivo versus in vitro, however, represents a more likely explanation for this failure. In conclusion, altered behavioral responses to morphine challenge in *Gpr88* null mice are consistent with an inhibitory influence of GPR88 co-expression on μOR signaling, especially for behaviors engaging striatal regions, thus demonstrating the predictive value of in vitro experiments.

Further confirming such predictive value, in vivo administration of Compound 19 markedly inhibited morphine-induced stimulation of locomotor activity in wild-type mice, resulting in opposed effects compared to *Gpr88* deletion on this parameter. GPR88 activation, however, had no influence on the amplitude of locomotor sensitization measured upon repeated morphine administration. Interestingly, while acute stimulant effects of morphine were shown to involve both G protein activation and β-arrestin recruitment by μOR (and D1 receptors), sensitization, as mentioned above, relies essentially on the latter (*Tao et al., 2017*; *Urs et al., 2011*; *Girault et al., 2007*). In light of Compound 19's effects on morphine-induced locomotor responses, it thus seems that pharmacological activation of GPR88 in vivo can potentiate the inhibitory effects of GPR88 on μOR-dependent G protein activation, but not β-arrestin recruitment, in agreement with in vitro results (*Figure 2—figure supplement 2*). Whether this applies to other GPCR partners of GPR88, however, would deserve further investigation. Of note, these results further argue for the inhibitory effects of GPR88 on the G-protein dependent signaling of a partner GPCR requiring GPR88 to be in an active-like conformation, as previously described for many interacting GPCRs (*Vilardaga et al., 2008*; *Goudet et al., 2005*; *Pin et al., 2019*).

A second major finding from this study is that GPR88 can bias the signaling of multiple GPCRs beyond opioid receptors. In vitro, we detected physical proximity between GPR88 and muscarinic M1 and M4, dopamine D2, adenosine $A_{2A}$ and GPR12 receptors, suggesting that GPR88 can form hetero-oligomers with them. Such close interactions are plausible in vivo as all these partner receptors share a common enriched expression in the striatum and extended amygdala, notably in medium spiny GABAergic neurons (*Ferré et al., 2007*; *Surmeier et al., 2007*; *Pellissier et al., 2018*; *Ignatov et al., 2003*), where GPR88 is also expressed. In contrast, we could not evidence proximity between GPR88 and dopamine D1 receptor, another GPCR with enriched striatal expression, and two GPCRs not known to be expressed in CNS neurons, the vasopressin V2R and chemokine CXCR4 receptors. Interestingly, these last results suggest that GPR88 forms close associations with other GPCRs in a selective manner. Further studies will aim at determining the molecular interface involved in interactions between GPR88 and other GPCRs.

At functional level, GPR88 dampened agonist-induced G protein pathway activation of most receptors to which it comes close, with the exception of adenosine $A_{2A}$ receptors. These in vitro effects are consistent with our previous observation of facilitated δOR, μOR and muscarinic receptor agonist-induced [$^{35}$S]-GTPγS binding in striatal membranes from *Gpr88*[-/-] mice (*Meirsman et al., 2016a*). Regarding G protein dependent pathway, therefore, close proximity appears necessary,

although not sufficient, to predict an inhibitory effect of GPR88. This result suggests that GPR88 exerts its influence on G protein recruitment through hetero-oligomerization, either by triggering conformational changes of the target partner less favorable to G protein coupling, as proposed for the angiotensin II AT$_2$ receptor (*AbdAlla et al., 2001*), or by steric hindrance of this coupling, as shown for the orphan receptor GPR50 (*Levoye et al., 2006a*). Interestingly, GPR88 displays an atypically long third intracellular loop, which may impede G protein recruitment at partner GPCRs. Focusing now on β-arrestins, our in vitro work evidenced the ability of GPR88 to hinder their recruitment by all the GPCRs that we examined in the present study, independently from its physical proximity to these target receptors. Interestingly, previous reports have evidenced the ability of two GPCRs, the vasopressin V2R and neurokinin NK1 receptors, upon activation by a selective agonist, to sequester β-arrestins in endosomes and dampen their recruitment by a partner GPCR (*Klein et al., 2001*; *Schmidlin et al., 2002*). GPR88 fulfills two conditions required to exert a similar effect on β-arrestin trafficking, under basal conditions: it demonstrates high affinity for β-arrestins (*Figure 6D*) and displays, besides localization at cell membrane, a substantial, patchy, intracellular expression (*Massart et al., 2016*), suggesting its presence in vesicular compartments, where it colocalizes with β-arrestins (*Figure 6E*). Interestingly, the failure of high levels of μOR expression to rescue μOR-mediated β-arrestin recruitment in presence of GPR88 is also consistent with GPR88 maintaining an intracellular pool of β-arrestins out of reach of its partner GPCRs (*Figure 2D*). Thus, a likely mechanism by which GPR88 inhibits the recruitment of β-arrestins at other GPCRs is by sequestering the formers in intracellular compartments, and this effect appears to affect GPCRs in a non-selective manner, as we could not find a co-expressed receptor for which it was not observed. Of note, CXCR4, which shares with GPR88 a high basal affinity for β-arrestins (*Figure 6D*) but shows a higher ratio of membrane versus intracellular expression (*Watts et al., 2013*), was not able to interfere with agonist-induced β-arrestin recruitment at μOR (*Figure 6A*), further highlighting the peculiarity of GPR88 action.

Experimental evidence supports the physiological relevance of previous in vitro findings. Indeed, mice lacking GPR88 were shown to display a diminished locomotor response to the stimulant effects of the dopamine D1 agonist SKF81297 (*Quintana et al., 2012*), matching with the increase in G protein dependent activation of D1 receptors that we detected in presence of GPR88. Reduced D1 receptor activation in the striatum of *Gpr88*$^{-/-}$ animals may have contributed to decreased alcohol reward in a CPP paradigm (*Ben Hamida et al., 2018*; *Cole et al., 2018*), not observed for morphine CPP as it was possibly compensated by facilitated μOR signaling, and reduced foraging efficiency in these animals (*Rainwater et al., 2017*). Of note and as mentioned above, increased β-arrestin recruitment at D1 receptor likely played a crucial role in the facilitation of morphine-induced locomotor sensitization that we observed in *Gpr88* null mice. Moreover, in these animals, the effects of D2 receptor activation on striatum-dependent locomotion, stereotypic behavior and catalepsy were shown to be exacerbated (*Quintana et al., 2012*; *Logue et al., 2009*), consistent with decreased signaling when GPR88 is co-expressed with the D2 receptor in vitro. Thus in vitro evidence that GPR88 interferes with dopamine receptor activity accurately predict pharmacologically-induced behavioral responses in vivo. More interestingly, these data further argue for a critical role of GPR88 in modulating the physiology of dopaminoceptive neuronal populations, primarily in the striatum and amygdala (*Meirsman et al., 2016a*; *Ben Hamida et al., 2018*; *Quintana et al., 2012*), where it would dampen GPCR activity. As a corollary of this, induction of *Gpr88* expression may represent an adaptive modulatory mechanism in these populations, in accordance with the propensity of psychoactive drugs targeting dopaminoceptive substrates to stimulate *Gpr88* transcription (*Conti et al., 2007*; *Ogden et al., 2004*; *Brandish et al., 2005*; *Le Merrer et al., 2012*; *Becker et al., 2017*). GPR88 could thus play a protective buffering role whenever striatal GPCRs are excessively activated, a role that may be lost or compromised in neuropsychiatric conditions (*Alkufri et al., 2016*; *Del Zompo et al., 2014*; *Ben Hamida et al., 2018*).

In conclusion, the orphan receptor GPR88 dampens the signaling of multiple GPCRs, the consequences of such inhibition depending on their location in the CNS and the signaling pathways involved. This study further highlights the interest of GPR88 as a promising target to treat various CNS disorders, together with the complexity of its pharmacology. Future investigations might consider developing, besides ligands modulating GPR88 activity, novel compounds able to interfere with its ability to closely interact with other GPCRs, or to sequester β-arrestins, in order to influence opioid function and/or striatal physiology.

# Materials and methods

## Key resources table

| Reagent type (species) or resource information | Designation | Source or reference | Identifiers | Additional |
|---|---|---|---|---|
| Chemical compound, drug | Arginine-Vasopressin (AVP) | Tocris Bioscience, Bristol, UK | Cat# 2935/1 | |
| Chemical compound, drug | Carbamoylcholine chloride | Tocris Bioscience, Bristol, UK | Cat# 2810/100 | |
| Chemical compound, drug | CGS21680 | Tocris Bioscience, Bristol, UK | Cat# 1063/10 | |
| Chemical compound, drug | Coelenterazine H substrate | Interchim, Montluçon, France | Cat# R30783 | |
| Chemical compound, drug | Compound 19 | Kindly synthetized by Domain Therapeutics *Dzierba et al., 2015* | | |
| Chemical compound, drug | DAMGO | Tocris Bioscience, Bristol, UK | Cat# 1171/1 | |
| Chemical compound, drug | Forskolin | Tocris Bioscience, Bristol, UK | Cat# 1099 | |
| Chemical compound, drug | IBMX | Tocris Bioscience, Bristol, UK | Cat# 2845/50 | |
| Chemical compound, drug | Ionomycin calcium salt | Tocris Bioscience, Bristol, UK | Cat# 1704/1 | |
| Chemical compound, drug | Metafectene PRO | Biontex, München, Germany | Cat#T040-5.0 | |
| Chemical compound, drug | Morphine HCl | Francopia, Antony, France | | |
| Chemical compound, drug | Naloxone | Sigma-Aldrich, Saint-Quentin Fallavier, France | Cat# N7758 | |
| Chemical compound, drug | Pertussis Toxin | Tocris Bioscience, Bristol, UK | Cat# 3097/50U | |
| Chemical compound, drug | Quinpirole hydrochloride | Tocris Bioscience, Bristol, UK | Cat# 1061/10 | |
| Chemical compound, drug | SKF81297 | Tocris Bioscience, Bristol, UK | Cat# 1447/10 | |
| Chemical compound, drug | SNC80 | Tocris Bioscience, Bristol, UK | Cat# 0764/10 | |
| Chemical compound, drug | Stromal cell-derived factor 1 (SDF-1) | Tocris Bioscience, Bristol, UK | Cat# 3951 | |

*Continued on next page*

*Continued*

| Reagent type (species) or resource information | Designation | Source or reference | Identifiers | Additional |
|---|---|---|---|---|
| Chemical compound, drug | U50488H | Tocris Bioscience, Bristol, UK | Cat# 0495/25 | |
| Material | Mithras$^2$ LB 943 Monochromator Multimode Microplate Reader | Berthold Technologies GmbH and Co. KG, Bad Wildbad, Germany | | |
| Material | MACSQuant10 flow cytometer | Miltenyi, Bergisch Gladbach, Germany | | |
| Material | Trans-Blot Turbo Transfer System | Bio-Rad, Hercules, California, USA. | | |
| Material | Odyssey CLx | LI-COR, Lincoln, Nebraska, USA | | |
| Material | LSM 700 laser scanning confocal microscope | Zeiss, Oberkochen, Germany | | |
| Material | Infrared floor | Videotrack; View Point, Lyon, France | | |
| Material | Computerized CPP boxes | Imetronic, Pessac, France | | |
| Material | Hot plate | Ugo Basile, Gemonio, Italia | | |
| Antibody | GAPDH (14C10) Rabbit mAb | Cell Signaling, Leiden, Netherlands | Cat# 2118 | (1:2000) |
| Antibody | Phospho-p44/42 MAPK (Erk1/2) (Thr202/Tyr204) (D13.14.4E) XP Rabbit mAb | Cell Signaling, Leiden, Netherlands | Cat# 4370 | (1:2000) |
| Antibody | p44/42 MAPK (Erk1/2) (3A7) Mouse mAb | Cell Signaling, Leiden, Netherlands | Cat# 9107 | (1:2000) |
| Antibody | Goat anti-rabbit IRDye800CW | LI-COR, Lincoln, Nebraska, USA | Cat# 926–3221 | (1:15000) |
| Antibody | Goat Anti-mouse IRDye680CW | LI-COR, Lincoln, Nebraska, USA | Cat# 926–68070 | (1:15000) |
| Antibody | Goat polyclonal anti-HA HRP-conjugated antibody | Bethyl Laboratories, USA | Cat#A190-138P | (1:10000) |
| Antibody | Anti-HA tag antibody - ChIP Grade | Abcam, Cambridge, UK | Cat# ab9110 | (1:300) |
| Antibody | Cy3 AffiniPure Donkey Anti-Rabbit IgG (H+L) | Jackson ImmunoResearch Europe Ltd, Cambridgeshire, Uk | Cat# 711-165-152 | (1:300) |
| Cell line (Human) | HEK 293FT cell line | ThermoFisher Scientific Inc, Waltham, Massachusetts, USA | Cat# R70007 | |

*Continued on next page*

*Continued*

| Reagent type (species) or resource information | Designation | Source or reference | Identifiers | Additional |
|---|---|---|---|---|
| Commercial assay or kit | SuperSignal ELISA Femto Maximum Sensitivity Substrate | ThermoFisher Scientific Inc, Waltham, Massachusetts, USA | | (1:10) |
| Commercial assay or kit | Trans-Blot Turbo RTA Midi Nitrocellulose Transfer Kit | Bio-Rad, Hercules, California, USA | Cat# 1704271 | |
| Commercial assay or kit | DAPI Hardset mounting medium | Vectashield Vector laboratories, Burlingame, California, USA | Cat# H-1500 | |
| Strain, strain background (*Mus musculus*) | *Gpr88*$^{+/+}$ and *Gpr88*$^{-/-}$, hybrid 50% C57BL/6J–50% 129Sv genetic background | **Meirsman et al., 2016a** | | |
| Strain, strain background (*Mus musculus*) | C57BL/6JRj | Janvier Labs, Le Genest-Saint-Isle, France | | |

## Plasmids

Plasmid encoding the different human receptor cDNAs, *GPR88* (GPR88), *OPRK1* (κOR), *OPRM1* (μOR), *OPRD1* (δOR), *ADORA2A* (A$_{2A}$), *DRD1* (D1), *DRD2* (D2), *GPR12* (GPR12), *CHRM1* (M1), *CHRM4* (M4) were purchased at Missouri S and T cDNA resource center, USA. All the receptors were tagged at their C-terminus with either Venus or Rluc8 BRET partners in pcDNA3. *CXCR4*-Venus, *CXCR4*-Rluc8, *AVPR2*-Rluc8 (V$_2$R), *β-arrestin1*-Venus, *β-arrestin2*-Venus, YPet-*β-Arrestin2*, *Rab5*-Venus, *Rab7*-Venus, *Kras*-Venus, CAMYEL (cAMP sensor using YFP-Epac-Rluc), Aequorin-GFP (calcium-sensitive photoprotein) constructs in pcDNA3/pcDNA3.1 were generous gifts from MA Ayoub (**Ayoub et al., 2013**). μOR, δOR, κOR and GPR88 were tagged at the N-terminus with mGluR5 signal peptide and the hemagglutinin (HA) epitope tag (MYPYNVPNYA) in pcDNA3.1 expression vector (HA-μOR, HA-δOR, HA-κOR and HA-GPR88.

## Cell line

In vitro experiments in this study were performed using the HEK293FT cell line (ThermoFisher Scientific Inc, Waltham, Massachussets, USA) that received the RRID CVCL_6911. Test for mycoplasma revealed no contamination.

## Chemical and drugs

DAMGO, U50488H, SNC80, CGS21680, AVP (arginine vasopressin), IBMX, Forskolin, stromal cell-derived factor 1 (SDF-1 or CXCL12), carbamoylcholine chloride (carbachol), quinpirole hydrochloride, SKF 81297, ionomycin calcium salt, were purchased from Tocris Bioscience (Bristol, UK) and diluted in DMSO (diméthylsulfoxyde) at $10^{-2}$ M (except SDF1 at 125 μM and IBMX at 200 mM) for frozen stock aliquots and coelenterazine H substrate from Interchim (Montluçon, France) was diluted in 100% ethanol and kept at $-20°$C. Protease/Phosphatase Inhibitor Cocktail were purchased from Cell Signaling Technology (Leiden, Netherlands). Pertussis toxin (PTX), purchased from Tocris, were diluted in water at 0.1 μg/μl and stored at 4°C. Morphine HCl was purchased from Francopia (Paris, France). Phenylmethanesulfonyl fluoride (PMSF) was diluted in isopropyl alcohol at 200 mM and stored at $-20°$C. Compound 19 was generously synthetized by Domain Therapeutics (Illkirch, France). For in vitro studies, it was diluted in DMSO (diméthylsulfoxyde) at $10^{-2}$ M and frozen at $-20°$C. For in vivo studies, Compound 19 was kept as a powder at 4°C and diluted in a saline solution before ICV administration (NaCl 9%).

## Antibodies

In western blot experiments, primary antibodies targeting GAPDH (Rabbit mAb, ref 2118; RRID:AB_561053) (1:2000), phospho-p44/42 MAPK (ERK1/2) (Thr202/Tyr204) (Rabbit mAb, ref 4370; RRID: AB_2315112) (1:2000) and MAPK (Erk1/2) (mouse mAb, ref 9107; RRID:AB_10695739) (1:2000) from Cell Signaling Technology were used at the indicated dilutions. They were combined with the following secondary antibodies: goat anti rabbit IRDye800CW (LI-COR, USA, 926–3221; RRID AB_621843) (1:15000) and goat anti mouse IRDye680RD LI-COR (926–68070; RRID AB_10956588) (1:15000). For immunocytochemistry experiments, primary rabbit polyclonal anti-HA antibody (ab9110; RRID:AB_307019) (Abcam, UK; 1:300) was used with secondary anti-rabbit-cyanin3 secondary antibodies (Jackson ImmunoResearch, 711-165-152; RRID:AB_2307443) (1:1000).

## Cell culture and transfection

HEK293FT cells were grown in Dulbecco's modified Eagle's medium (DMEM) supplemented with 10% (v/v) foetal bovine serum, 100 U/ml penicillin, 0.1 mg/ml streptomycin, and 1 mM glutamine (Eurobio) at 37°C in 5% CO2. They were then transiently transfected using Metafectene PRO (Biontex) following the manufacturer's instructions.

## In vitro testing

### Oligomerization assays

Oligomerization assay was performed as described in *Borroto-Escuela et al. (2013)*. Briefly, $5.10^4$ HEK293FT cells were co-transfected with Rluc-tagged receptors (30 ng, except 50 ng for SP-μOR and 20 ng for δOR-RLuc8) and increasing amount of Venus-tagged receptors (0–100 ng) or YPet-β-arrestin2 (0–100 ng) in white 96-well plates (Greiner Bio-One). Each transfection was duplicated in black 96-well plates (Greiner Bio-One) in order to measure the expression level of the Venus-tagged receptor or YPet-β-arrestin2. 36 hr after transfection, cells were starved for 16 hr. In black plates, cells were incubated in PBS containing 5 mM HEPES and Venus or YPet fluorescence was measured at 540 nm (excitation 480 nm) using the Mithras LB 943 plate reader (Berthold). Net Venus or YPet fluorescence was calculated as sample fluorescence minus the fluorescence of the Rluc-only sample of each experiment. For BRET, white plates were incubated in PBS containing 5 mM HEPES and 5 μM coelenterazine H and emission at 480 nm (Rluc8 bioluminescence) and at 540 nm was immediately measured for 35 min using the Mithras LB 943 plate reader (Berthold). BRET ratios (540 nm emission/480 nm emission) were calculated and net BRET, ([BRET] - [BRET in absence of Venus acceptor]) expressed as BRET units (Bu) or percentage of maximal response, was plotted as a function of the net Venus fluorescence (acceptor)/Rluc8 luminescence (donor) ratio.

### cAMP accumulation assays

$5.10^4$ HEK293FT cells were co-transfected with untagged receptors, CAMYEL (50 ng) and eventually in presence of increasing amounts of GPR88 (0–75 ng) in 96-well plates. 36 hr after transfection, cells were starved for 16 hr and stimulated with agonists in PBS buffer containing 250 μM IBMX, 5 mM HEPES and 5 μM coelenterazine H in presence or absence of 5 μM forskolin. BRET was immediately measured for 35 min using the Mithras LB 943 plate reader (Berthold). Stable BRET ratios, between minutes 10 and 20, were averaged for calculation. Induced BRET is the difference between ligand-induced BRET and basal BRET. Percentage of maximal response (absence of GPR88) of induced BRET was plotted as a function of GPR88 cDNA transfected.

### β-arrestin recruitment assays

$5.10^4$ HEK293FT cells were co-transfected with Rluc-tagged receptors and YPet-β-arrestin2 (50–75 ng) eventually in presence of increasing amounts of GPR88 (0–75 ng) in white 96-well plates (Greiner Bio-One). Each transfection was duplicated in black 96-well plates (Greiner Bio-One) in order to measure the expression level of YPet-β-arrestin2. 36 hr after transfection, cells were starved for 16 hr. In black plates, cells were incubated in PBS containing 5 mM HEPES and YPet fluorescence was measured at 540 nm (excitation 480 nm) using the Mithras LB 943 plate reader (Berthold). Net YPet fluorescence was calculated as the sample fluorescence minus the fluorescence of the Rluc-only sample of each experiment. For BRET, white plates were stimulated with vehicle or agonists in PBS containing 5 mM HEPES and 5 μM coelenterazine H and emission at 480 nm (Rluc8 bioluminescence) and at

540 nm was immediately measured for 35 min using the Mithras LB 943 plate reader (Berthold). BRET ratios (540 nm emission/480 nm emission) were calculated and net BRET ([BRET] - [BRET in absence of Venus acceptor]) was normalized to the YPet fluorescence signal intensity. Ligand-induced BRET changes were calculated as the difference between basal and agonist values, expressed as a percentage of the maximal response (absence of GPR88) and plotted as a function of the amount of GPR88 cDNA transfected.

## Calcium mobilization assays

$1.10^6$ HEK293FT cells were co-transfected in a 6-well plate with plasmids coding for muscarinic M4 (60 ng), aequorin-GFP (250 ng) with increasing amounts of a plasmid coding for GPR88. 36 hr after transfection, cells were starved for 16 hr and harvested in calcium-free HBSS (Hanks Balanced Salt Solution, Eurobio) containing 5 µM of coelenterazine H for 3 hr at 37°C. They were then incubated in HBSS containing 5 µM of coelenterazine H for 1 hr at 37°C. Finally, 40 µl of the cell suspension ($1.10^6$ cells/ml) were injected in white 384-well plates containing 10 µl of carbachol (50 µM). Luminescence was read for 20 s using the Mithras LB 943 plate reader. To normalize the signal, 10 µl of ionomycin (100 µM) was injected in the same wells and luminescence was read for 30 s. Area under the curves (AUC) of both runs were calculated and ligand-mediated calcium signal was determined as AUC (carbachol)/[AUC (carbachol) + AUC (ionomycin)].

## Internalization assays

$5.10^4$ HEK293FT cells were co-transfected with Rluc-tagged receptors (30 ng), Kras-Venus (10 ng) or Rab5-Venus (30 ng) or Rab7-Venus (30 ng) and increasing amounts of GPR88 (0–75 ng) in white 96-well plates (Greiner Bio-One). Each transfection was duplicated in black 96-well plates (Greiner Bio-One) in order to measure the expression level of the Kras-Venus, Rab5-Venus or Rab7-Venus. 36 hr after transfection, cells were starved for 16 hr. In black plates, cells were incubated in PBS containing 5 mM HEPES and Venus fluorescence was measured at 540 nm (excitation 480 nm) using the Mithras LB 943 plate reader (Berthold). Net Venus fluorescence was calculated as sample fluorescence minus the fluorescence of the Rluc-only sample of each experiment. For BRET, cells were stimulated with agonists in PBS containing 5 mM HEPES and 5 µM coelenterazine H and emission at 480 nm (Rluc8 bioluminescence) and at 540 nm was immediately measured for 35 min using the Mithras LB 943 plate reader (Berthold). BRET ratios (540 nm emission/480 nm emission) were calculated and net BRET ([BRET] - [BRET in absence of Venus acceptor]) was normalized to the Venus fluorescence signal intensity. Percentage of maximal response of normalized net BRET was plotted as a function of GPR88 cDNA transfected.

## ERK phosphorylation assays (in vitro experiments)

$1.10^6$ HEK293FT cells were co-transfected either with SP-µOR (500 ng) or SP-µOR (500 ng) and GPR88 (1000 ng). 36 hr after transfection, cells were starved overnight in presence or absence of 0.1 ng/µl Pertussin toxin. They were then stimulated with 10 nM DAMGO for 0, 1, 5, 10, 15, 20, 30 and 60 min. After removing the supernatant, stimulation, proteases and phosphorylation were stopped by adding 150 µL of Sample Buffer (125 mM Tris, 20% glycerol, 4% SDS, 10% β-mercaptoethanol, pH = 6.8) containing 1X Protease/Phosphatase Inhibitor Cocktail. The samples were then frozen at −20°C.

For western blot analysis, samples were denatured at 95°C for 15 min and cleared by centrifugation at 11,000 g for 5 min. Supernatant was loaded onto SDS-acrylamide 4–15% Mini-PROTEAN TGX Precast Protein Gels (BioRad) and proteins were transferred to nitrocellulose membranes with Trans-Blot Turbo RTA transfer kit (BioRad) using the Trans-Blot Turbo Transfer System (BioRad). Membranes were blocked with 5% (w/v) milk powder diluted in TBS-T (Tris-buffered saline with 1% (v/v) Tween20) for 1 hr and then incubated at 4°C overnight with the primary antibody. Blots were finally incubated, in the dark, with fluorescent secondary antibodies, at room temperature. Revelation was performed using the infrared scanner Odyssey CLx (LI-COR Biotechnology). Quantification was performed using Image Studio software. The signal from each band of p-ERK and tERK was divided by the GAPDH signal of the corresponding track for normalization. The ratio pERK/tERK was expressed as a fold change versus vehicle condition.

### ERK phosphorylation assays (ex vivo experiments)

Mouse brains were dissected immediately after locomotion experiments and placed into a brain matrix (ASI Instruments, Warren, MI, USA). Caudate putamen (CPu), nucleus accumbens (NAc), and periaqueductal gray (PAG) were punched out. Tissues were immediately homogenized on RIPA buffer (Cell Signaling Technology) containing 1 mM of PMSF and 1X Protease/Phosphatase Inhibitor Cocktail, incubated 30 min on ice, centrifuged at 10000 g for 10 min at 4°C, and supernatants were prepared for western blotting by adding the appropriate volume of 4X Laemmli sample buffer (NuPAGE LDS Sample buffer NP0007, ThermoFisher Scientific) containing 10% β-mercaptoethanol.

### Immunocytochemistry

HEK293FT cells were grown on 24-well plates (Greiner) containing GelTrex (Gibco)-coated coverslips (50 000 cells per well). 24 hr after plating, cells were transfected with SP-HA-μOR with or without GPR88-Venus (20 ng each) or SP-HA-GPR88 with or without YPet-β-arrestin 2 (20ng each) using Metafectene PRO according manufacturer instructions. 36 hr after transfection, they were starved for 16 hr, stimulated or not with 10 μM of agonist for 30 min at 37°C (5% $CO_2$) and then fixed with 4% paraformaldehyde (Sigma-Aldrich) diluted in phosphate buffered saline (PBS) for 5 min. The immunocytochemical experiments were performed at room temperature. After a 5 min washing step (1% PBS-BSA), cells were permeabilized using 0.1% Triton X-100 diluted in 1% PBS-BSA for 5 min and then saturated for 1 hr with 1% PBS-BSA. Then, cells were incubated for 1 hr with rabbit poly-clonal anti-HA antibodies in 1% PBS-BSA. After 3 washing steps (1% PBS-BSA), anti-rabbit-cyanin3 secondary antibodies were applied for 1 hr. Cells were mounted using DAPI Hardset mounting medium (Vectashield Vector laboratories). Acquisitions were performed using a LSM 700 laser scanning confocal microscope (Zeiss), at 488 nm (Venus) and 568 nm (Cyanin3) and pictures were analyzed using Image J 1.5 (NIH).

## Cell surface protein expression

For ELISA experiments, cells were transfected with SP-HA-tagged GPCRs in Poly-L-lysine coated 96-well plates. 36 hr after transfection, cells were starved for 16 hr. PBS-BSA-1% buffer was used as blocking solution during 1 hr and cells were incubated in the dark during 1 hr with anti-HA HRP conjugated antibody (Bethyl Laboratories, USA, Cat#A190-138P; RRID:AB_2631897). Cells were washed 3 times with PBS-BSA-1% buffer before using SuperSignal ELISA Femto Maximum Sensitivity Substrate (Thermo scientific, France) on cells and luminescence acquisition was read three times using the Mithras LB 943 plate reader (Berthold, France). Values (luminescence arbitrary units) were averaged and normalized as a percentage of the value measured for the receptor expressed alone.

For cytometry experiments, cells were transfected with SP-HA-tagged GPCRs and fixed with 4% paraformaldehyde (Sigma, France) diluted in phosphate buffered saline (PBS) for 5 min, and permeabilized in triton 0,1%, in case of intracellular staining. All steps were performed at room temperature. After 1 hr saturation in PBS-0,1%BSA, cells were incubated with anti-HA antibody conjugated to the horse radish peroxidase (HRP) (Bethyl Laboratories, USA; 1:10,000) in PBS-0,1% BSA for 1 hr, in the dark, followed by 3 washes in PBS-0,1% BSA followed by 3 washes in PBS. Fluorescence was read using the 488 nm excitation laser and 565–605 nm emission filter of the MACSQuant10 flow cytometer (Myltenyi, Germany). Mean fluorescence in non-permeabilized condition and non-permeabilized/permeabilized ratio were normalized as percentage for cell surface and total protein expression of mutants versus wild-type receptors.

## Animals

Constitutive GPR88 knockout animals were generated by breeding $Gpr88^{fl/+}$ animals with a general CMV-Cre driver line. This led to germ-line deletion of $Gpr88$ exon 2 on a hybrid 50% C57BL/6J–50% 129Sv genetic background. Equivalent numbers of male and female $Gpr88^{fl/fl}$ x CMV-Cre$^{Tg/+}$ and $Gpr88^{+/+}$ x CMV-Cre$^{0/+}$ aged 8 to 10 weeks were bred in house and used as experimental ($Gpr88^{-/-}$ mice) and control ($Gpr88^{+/+}$) animals, respectively (*Meirsman et al., 2016a*). As regards testing of Compound 19 effects in vivo, equivalent numbers of male and female C57BL/6JRj mice were purchased from Janvier Labs (Le Genest-Saint-Isle, France). Animals were group-housed and maintained on a 12 hr light/dark cycle (lights on at 7:00 AM) at a controlled temperature (22 ± 2°C). Food and water were available ad libitum throughout the experiments, unless otherwise stated. Experiments

were analyzed blind to genotypes. All experimental procedures were conducted in accordance with the European Communities Council Directive 2010/63/EU and approved by the Comité d'Ethique pour l'Expérimentation Animale de l'ICS et de l'IGBMC (Com'Eth, 2012–047).

## Behavioral testing

### Five-day morphine-induced locomotor sensitization (Gpr88$^{+/+}$ versus Gpr88$^{-/-}$ mice)

Locomotor activity was assessed in clear Plexiglas boxes (21 × 11 × 17 cm) placed over a white Plexiglas infrared-lit platform. Light intensity of the room was set at 15 lx. The trajectories of the mice were analyzed and recorded via an automated tracking system equipped with an infrared-sensitive camera (Videotrack; View Point). To focus on forward activity, only movements which speed was over 6 cm/s were taken into account for the measure of locomotor activity. Behavioral testing started when the animals were placed in the activity boxes for a habituation period of 60 min (Gpr88$^{+/+}$ versus Gpr88$^{-/-}$ mice in *Figure 2A*) or 30 min (Compound19 in wild-type mice, *Figure 2F*; short habituation facilitates context-drug association). They were then injected with either vehicle or morphine (5 mg/kg), and locomotor activity was monitored for further 120 min. This experiment was repeated for 5 consecutive days in experiments comparing Gpr88$^{-/-}$ to Gpr88$^{+/+}$ mice (*Figure 2A*) or 3 consecutive days in experiments assessing the effects of Compound19 in wild-type mice (*Figure 2F*).

### Three-day morphine-induced locomotor sensitization (effects of Compound 19 in vivo)

Locomotor activity was assessed in clear Plexiglas boxes (21 × 11 × 17 cm) placed over a white Plexiglas infrared-lit platform. Light intensity of the room was set at 15 lx. The trajectories of the mice were analyzed and recorded via an automated tracking system equipped with an infrared-sensitive camera (Videotrack; View Point). To focus on forward activity, only movements which speed was over 6 cm/s were taken into account for the measure of locomotor activity. On days 1, 2 and 3, mice received an icv injection (*Haley and McCORMICK, 1957*, 5 µl) of vehicle (saline 0.9%) or Compound 19 (10 nmoles) in their home cage 15 min before being placed in the activity boxes for a 30 min-habituation period. They were then injected with either vehicle (saline 0.9%) or morphine (10 mg/kg), and locomotor activity was monitored for further 120 min. The 10 nmoles dose of Compound 19 was the maximal dose to be well tolerated in a pilot experiment (convulsing effects at higher doses). We chose a higher dose of morphine than for experiments in Gpr88$^{+/+}$ and Gpr88$^{-/-}$ mice (5 mg/kg) to ensure greater locomotor activation and sensitization.

### Two-day morphine-induced locomotor sensitization (followed by ERK phosphorylation assays)

Locomotor activity was assessed in clear Plexiglas boxes (21 × 11 × 17 cm) placed over a white Plexiglas infrared-lit platform. Light intensity of the room was set at 15 lx. The trajectories of the mice were analyzed and recorded via an automated tracking system equipped with an infrared-sensitive camera (Videotrack; View Point). To focus on forward activity, only movements which speed was over 6 cm/s were taken into account for the measure of locomotor activity. On day 1, behavioral testing started when the animals were placed in the activity boxes for a 30 min-habituation period. They were then injected with either vehicle or morphine (5 mg/kg), and locomotor activity was monitored for further 120 min. On day 2, mice were recorded for a 30 min-habituation period before receiving vehicle or morphine (5 mg/kg) injection; locomotor activity was monitored for further 60 min. Mice were sacrificed within 5 min after the end of this behavioral session; brains were removed and placed into a brain matrix (ASI Instruments, Warren, MI, USA). Caudate putamen (CPu), nucleus accumbens (NAc) and periacqueductal gray were punched out from 1mm-thick slices. Tissues were immediately homogenized on RIPA buffer (Cell Signaling Technology) containing 1 mM of PMSF, incubated 30 min on ice, centrifuged at 10000 g for 10 min at 4°C, and supernatants were prepared for western blotting by adding the appropriate volume of 4X Laemmli sample buffer (NuPAGE LDS Sample buffer NP0007, ThermoFisher Scientific) containing 10% β-mercaptoethanol.

## Morphine withdrawal

Animals received daily morphine (30 mg/kg, s.c,) or saline injection for 5 consecutive days. On day 6, naloxone at 1 mg/kg (s.c.) was injected 2 hr after morphine or saline administration. Naloxone-induced withdrawal signs were then scored for 20 min. The number of head shakes and wet dog shakes, front paw tremors, jumps and sniffing episodes was counted. Vegetative signs of withdrawal, namely ptosis, mastication, and piloerection, were scored 1 for appearance or 0 for nonappearance within 5 min bins. A global withdrawal score was calculated for each animal by giving each somatic sign a relative weight, following the formula: (paw tremors x 0.35) + (wet dog shakes x 1) + (jumps x 0.8) + (sniffing episodes x 0.5) + (vegetative signs x 1.5) (adapted from *Berrendero et al., 2002*; *Matthes et al., 1996*).

## Conditioned place preference

Place conditioning experiments were performed in unbiased computerized boxes (Imetronic) formed by two Plexiglas chambers (15.5 × 16.5 × 20 cm) separated by a central alley (6 × 16.5 × 20 cm). Two sliding doors (3 × 20 cm) connected the alley with the chambers. Two triangular prisms of transparent polycarbonate were arranged in one chamber, and one rectangular prism in the other to form different shape patterns (covering the same surface). Distinct textured removable floors made of translucent polycarbonate provided additional contextual cues. The activity and location of mice were recorded using five photocells located throughout the apparatus. Behavioral data were collected by an interface connected to a PC. Light intensity in the chambers was set at 30 Lx.

Morphine conditioning consisted of 3 phases. On day 1, naïve mice were placed in the central alley and allowed to freely explore the apparatus for 20 min. Based on the individuals' spontaneous preference during this pretest phase, the drug-paired chambers were assigned in such a way that saline and morphine groups were counterbalanced and unbiased towards contextual cues. Conditioning phase lasted 3 days. Mice underwent two conditioning, vehicle and drug-paired, sessions daily, 7 hr apart. During vehicle and drug-conditioning sessions, animals were confined immediately after s.c. injection of morphine or vehicle for 45 min in the appropriate drug or vehicle-paired chamber. The testing phase was conducted on day 5, at the same time of the day as the pretest session. The animals, in a drug-free state, were placed in the neutral central alley and permitted to explore the apparatus for 20 min with the two sliding doors opened. The time spent in each chamber was recorded. All the animals tested were naïve when conditioning started.

The extinction phase was conducted during 10 constitutive days. The animals, in a drug-free, were placed in the neutral central alley and permitted to explore the apparatus for 20 min with the two sliding doors opened. Then the animal were injected with either morphine (10 mg/kg; s.c.) or saline solution and were placed in the neutral central alley of the apparatus for a 20 min reinstatement phase, during which they were permitted to explore freely the apparatus (sliding doors opened).

## Measure of nociception

Tail immersion test was performed using water at 48°C, 50°C and 52°C as the nociceptive stimulus. The mice were maintained in a cylinder and their tails were immersed in the heated water. The latency to a rapid flick of the tail was taken as the endpoint, and the maximum latency allowed was 10 s. The hot plate (Ugo Basile) surface was kept at 52 ± 0.1°C. A glass cylinder (16 cm of diameter) was used to maintain the mice on the plate. The licking and jumping responses were measured, and a 300 s cut-off was set to prevent tissue damage. Morphine (5 mg/kg) or saline was administered 20 min before performing the test.

## Data analysis and statistics

For BRET experiments, Rluc8 and Venus expression levels were measured separately, at 480 nm and 540 nm, respectively. The 540 nm/480 nm BRET ratio was plotted over 30 min and the area under the curve (AuC) was calculated. Net BRET (BRET-BRET in absence of Venus acceptor) was calculated and normalized by Venus/Rluc8 ratio. Induced BRET was calculated as the difference between ligand-induced BRET and basal BRET for each receptor transfected. Maximal response of ligand-induced BRET (in absence of GPR88) was considered as 100% of the receptor response and ligand-induced BRET in presence of increasing amounts of GPR88 cDNA was expressed as a percentage of

this maximal response. For experiments assessing the effects of increasing μOR amounts on GPR88 inhibitory effects, ligand-induced BRET in presence of increasing amounts of μOR cDNA was expressed as a percentage of maximal response for DAMGO-induced cAMP inhibition (*Figure 2C*) or a percentage of DAMGO-induced β-arrestin recruitment in the absence of GPR88 (*Figure 2D*). For saturation experiments (normalized as percentage), nonlinear hyperbola or linear fit regression (best fit) was used and analyzed using Prism. Obtained $BRET_{max}$ values significantly different from 100 percent were not considered saturated.

Statistical analyses were performed using Statistica 9.0 software (StatSoft, Paris). For all comparisons, values of $p < 0.05$ were considered as significant. For in vitro pharmacology experiments, statistical analysis was performed using non parametric Kruskal-Wallis ANOVA for comparison between GPR88 cDNA doses, followed by multiple comparison of mean ranks. For western blotting experiments, comparison between conditions (MOR versus MOR + GPR88) were performed at each time point using Kruskal-Wallis ANOVA for comparison between conditions (MOR versus MOR+GPR88), followed by comparison of mean ranks. For in vitro experiments testing the effects of transfecting increasing amounts of μOR, statistical significance was assessed using one-way analysis of variance (GPR88 effect) with a repeated measure (μOR effect). For in vivo experiments, statistical significance was assessed using one- to two-way analysis of variance (Genotype and Treatment effects) eventually with a repeated measure (time course) followed by Newman-Keules post-hoc test.

## Acknowledgements

The authors thank Dr M Darmon, Dr F Simonin, Dr M A Ayoub, T Boulo, G Tsikis and M-C. Blache for advises on experimental protocols, technical assistance and materials. We thank Dr J Kniazeff and Dr P Crépieux for critical reading of the manuscript. This study has benefited from the facilities and expertise of the Plateforme d'Imagerie Cellulaire (PIC) of the Laboratory Physiologie de la Reproduction et des Comportements. This work was funded by the Institut National de Recherche pour l'Agriculture, l'Alimentation et l'Environnement (INRAE), Centre National de la Recherche Scientifique (CNRS), Institut National de la Santé et de la Recherche Médicale (Inserm), and Université de Tours. We thank Région Centre-Val de Loire (ARD2020 Biomédicaments – GPCRAb), LabEx MAbImprove and the ATHOS Consortium, including the Fonds Unique Interministériel (FUI), the Région Alsace and our partners, Domain Therapeutics (Illkirch, France) and Prestwick Chemicals (Illkirch, France), for support in this project. LP Pellissier acknowledges postdoctoral funding from the Marie-Curie/AgreenSkills Program.

## Additional information

### Funding

| Funder | Grant reference number | Author |
|---|---|---|
| Region Centre-Val de Loire | ARD2020 Biomedicaments - GPCRAb | Julie Le Merrer<br>Jérôme AJ Becker |
| LabEX MAbImprove | | Julie Le Merrer<br>Jérôme AJ Becker |
| Fonds Unique Interministériel | ATHOS | Brigitte L Kieffer<br>Julie Le Merrer<br>Jérôme AJ Becker |
| H2020 Marie Skłodowska-Curie Actions | AgreenSkills Program Postdoctoral Fellowship | Lucie P Pellissier |
| Institut National de Recherche pour l'Agriculture, l'Alimentation et l'Environnement | | Thibaut Laboute<br>Jorge Gandía<br>Lucie P Pellissier<br>Yannick Corde<br>Christophe Gauthier<br>Anne Poupon<br>Julie Le Merrer<br>Jérôme AJ Becker |

| Centre National de la Recherche Scientifique | Jérôme AJ Becker |
|---|---|
| Inserm | Jérôme AJ Becker |
| Université de Tours | Jérôme AJ Becker |

The funders had no role in study design, data collection and interpretation, or the decision to submit the work for publication.

## Author contributions
Thibaut Laboute, Conceptualization, Resources, Formal analysis, Validation, Investigation, Visualization, Methodology, Writing - original draft, Writing - review and editing; Jorge Gandía, Conceptualization, Resources, Formal analysis, Validation, Investigation, Visualization, Methodology, Writing - review and editing; Lucie P Pellissier, Conceptualization, Formal analysis, Investigation, Visualization, Methodology, Writing - original draft; Yannick Corde, Formal analysis, Validation, Investigation, Visualization; Florian Rebeillard, Maria Gallo, Christophe Gauthier, Resources, Investigation; Audrey Léauté, Investigation; Jorge Diaz, Resources, Investigation, Writing - review and editing; Anne Poupon, Resources; Brigitte L Kieffer, Conceptualization, Resources, Formal analysis, Investigation, Methodology, Writing - review and editing; Julie Le Merrer, Jérôme AJ Becker, Conceptualization, Resources, Formal analysis, Supervision, Funding acquisition, Validation, Investigation, Visualization, Methodology, Writing - original draft, Project administration, Writing - review and editing

## Author ORCIDs
Thibaut Laboute ⓘ https://orcid.org/0000-0003-0870-1891
Jorge Gandía ⓘ https://orcid.org/0000-0003-1711-8075
Brigitte L Kieffer ⓘ http://orcid.org/0000-0002-8809-8334
Julie Le Merrer ⓘ https://orcid.org/0000-0002-8670-4273
Jérôme AJ Becker ⓘ https://orcid.org/0000-0002-0039-0067

## Ethics
Animal experimentation: All experimental procedures were conducted in accordance with the European Communities Council Directive 2010/63/EU and approved by the Comité d'Ethique pour l'Expérimentation Animale de l'ICS et de l'IGBMC (Com'Eth, 2012-047).

## Decision letter and Author response
Decision letter https://doi.org/10.7554/eLife.50519.sa1
Author response https://doi.org/10.7554/eLife.50519.sa2

# Additional files
## Supplementary files
• Supplementary file 1. Statistical analysis of western blotting data from in vitro experiments (Kruskal-Wallis ANOVA).

• Transparent reporting form

## Data availability
All data generated or analysed during this study are included in the manuscript and supporting files.

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
