## [Decision Letter]

**Acceptance summary:**

GPR88 is an orphan GPCR (its endogenous ligand is not known) mainly expressed in the striatum and already shown to play important roles in brain physiology. The authors previously characterized the GPR88 knockout mice and show that the functional responses of the M1 muscarinic, mu and delta opioid receptors were potentiated. In the present study, the authors aimed at clarifying how GPR88 can potentiate these responses in transfected cells, and analyze the behavioral consequences of such an effect in mice. The results lead to two major observations. First, they show that GPR88 inhibits G-protein-mediated responses of various GPCRs via its likely association with the targeted receptor, such that only the coupling of those receptors interacting with GPR88 is affected, while GPR88 dampened β-arrestin recruitment by all GPCR tested likely through a sequestering mechanism, then affecting β-arrestin-mediated signaling of all GPCR tested. Such a mechanism then leads to the signaling bias of those receptors not interacting with GPR88 as only their β-arrestin signaling is affected. The second important finding is that only some of the opioid receptor mediated behavioral responses are potentiated in GPR88 knockout mice, consistent with those responses involving brain structures in which GPR88 is normally expressed. These findings are of major interest as they extend our view of the possible actions of 7TM membrane proteins.

**Decision letter after peer review:**

Thank you for submitting your article "The orphan receptor GPR88 blunts the signaling of opioid receptors and multiple striatal GPCRs" for consideration by *eLife*. Your article has been reviewed by three peer reviewers, including Volker Dötsch as the Reviewing Editor and Reviewer #1, and the evaluation has been overseen by Richard Aldrich as the Senior Editor. The following individual involved in review of your submission has agreed to reveal their identity: Jean-Philippe Pin (Reviewer #2).

The reviewers have discussed the reviews with one another and the Reviewing Editor has drafted this decision to help you prepare a revised submission.

Essential revisions:

1) Quantification of the cell surface receptors is key in this study, as it is required to conclude that the effects observed are due to a decrease in coupling, rather than a decrease in surface expression of the receptor. Indeed, very often, when a second protein is overexpressed in the same transfected cells, a decrease in the expression of the first protein is observed due to the limit in the translation machinery. The authors estimate the amount of cell surface proteins by ELISA, taking advantage of an N-terminal HA tag. ELISA can easily saturate leading to a similar signal despite a possible large difference in expression level. The authors must document that they are in the linear range of they ELISA assay under their conditions, and that indeed, higher expression level lead to higher ELISA signal. Such data must be added in any situation where cell surface receptors are quantified.

2) GPR88 is shown to have differential relative inhibitory effect on different receptors. This may likely be due to a difference in the receptive expression of the receptor compared to GPR88. Indeed, if the inhibitory action needs direct interaction between the receptors, GPR88 may not be as efficient in inhibiting a receptor that is expressed as a higher level than GPR88, compare to a receptor that is less expressed than GPR88. This should be analyzed. Similarly, looking at the coupling ability of increasing target receptors with a fixed amount of GPR88 would be informative especially if the relative expression of both could be estimated. In other words, can one GPR88 inhibit only one of its target, or several of them?

3) To be considered as an interesting drug target, as mentioned several times by the authors, it would be necessary to show that the effect reported here on GPR88 can be indeed modulated by drugs. It would be nice if the authors could test at least one of the synthetic ligand reported to active GPR88. Of course, testing an inhibitor would also be a nice addition but as far as I am aware, no such compound is available yet. Alternatively, analyzing the effect of a constitutively active mutant of GPR88, or a loss of function mutation of this receptor would be nice.

4) It become more and more clear that interacting 7TM receptors control each other through direct interaction, and in most cases, the activation of one, prevents the activation of its partner (see the elegant work of M Lohse on this topic using FRET base GPCR sensors (Vilardaga et al., 2008), also supported by many data from the Pin's group (see their recent review on this topic in MCE 2019). As such, GPR88 could inhibit the coupling of a GPCR partner if in an active-like conformation. Can this be tested? This should be mentioned as a possibility at least.

5) The pERK signal appears to be very weak, which suggest potential dephosphorylation during sample handling. How were the data normalized?

6) The authors should consider including siRNA experiments to establish that the second wave of Erk activation is arrestin dependent. In the present stage the only thing that can be concluded is that this later phase of signaling is more resistant to PTX. Also, there is a non-replicability at the 30 min time point if data are explained solely by PTX resistant signaling at later phases.

---

## [Author Response]

Essential revisions:1) Quantification of the cell surface receptors is key in this study, as it is required to conclude that the effects observed are due to a decrease in coupling, rather than a decrease in surface expression of the receptor. Indeed, very often, when a second protein is overexpressed in the same transfected cells, a decrease in the expression of the first protein is observed due to the limit in the translation machinery. The authors estimate the amount of cell surface proteins by ELISA, taking advantage of an N-terminal HA tag. ELISA can easily saturate leading to a similar signal despite a possible large difference in expression level. The authors must document that they are in the linear range of they ELISA assay under their conditions, and that indeed, higher expression level lead to higher ELISA signal. Such data must be added in any situation where cell surface receptors are quantified.

To quantify cell surface expression of opioid receptors in the conditions of Figure 2, we used a goat polyclonal anti-HA HRP-conjugated antibody (Bethyl Laboratories, USA; 1:10,000 as recommended by the manufacturer). To address the reviewer’s question, we performed 3 series of experiments: 1) we measured the signal (luminescence) emitted by the antibody (1:10,000) in presence of increasing amounts of cells expressing either HA-tagged δOR, κOR or µOR (1 µg cDNA for 1,000,000 cells); 2) we measured the signal emitted by the antibody (1:10,000) in presence of a fixed amount of cells transfected with increasing amounts of HA-tagged µOR cDNA (µOR was chosen for being the most difficult among opioid receptors to express in HEK cells); 3) we measured the signal emitted by increasing concentrations of the antibody in presence of a fixed amount of µOR cDNA (30 ng).

As now shown in Figure 1—figure supplement 2A, in the first experiment (panel a), the antibody displayed a linear response to increasing amounts of HA-tagged GPCR-expressing cells in the 50,000 to 200,000 range. Thus, testing conditions (black arrow) in Figure 2 were satisfying to detect surface expression of opioid receptors (n=2-4 per receptor, in triplicates). This was further demonstrated for µOR in the second experiment (panel b). When the cell number was set at 50,000, the anti-HA HRP-conjugated antibody allowed reliable detection of increasing amounts of transfected HA-tagged µOR (n=4, in triplicates). Cells in Figure 2 were transfected with 30 ng cDNA (black arrow). Finally, in a third experiment (panel c), we reveal that the 1:10,000 concentration of antibody chosen for our experiments (black arrow; the manufacturer recommends 1:10,000 to 1:100,000) allows optimal detection of HA-µOR before saturating for higher concentrations (over 1:5,000) (n=1, in triplicates). Taken together, these complementary experiments indicate that, under our conditions, we were indeed in the linear range of the ELISA assay, allowing us to satisfyingly detect modifications in the cell surface expression of opioid receptors.

2) GPR88 is shown to have differential relative inhibitory effect on different receptors. This may likely be due to a difference in the receptive expression of the receptor compared to GPR88. Indeed, if the inhibitory action needs direct interaction between the receptors, GPR88 may not be as efficient in inhibiting a receptor that is expressed as a higher level than GPR88, compare to a receptor that is less expressed than GPR88. This should be analyzed. Similarly, looking at the coupling ability of increasing target receptors with a fixed amount of GPR88 would be informative especially if the relative expression of both could be estimated. In other words, can one GPR88 inhibit only one of its target, or several of them?

To address this question, we focused on µOR and evaluated the effects of a fixed amount of co-expressed GPR88 on the ability of increasing amounts of µOR to inhibit cAMP production or β-arrestin recruitment. These experiments are now presented in Figure 2C.

As regards G-protein dependent signaling, we observed that the inhibitory effect of GPR88 on µOR signaling weakens when amounts of µOR increase, arguing for a direct interaction between the two receptors. Interestingly, raising the amount of GPR88 expressed (from 30ng of GPR88 cDNA transfected to 50ng) shifted this curve to the right, confirming our previous results shown in Figure 1 (30ng of µOR cDNA, framed on Figure 2C). Of note, however, at these levels of GPR88 expression (30 or 50 ng of cDNA), increasing µOR expression allowed to completely overcome GPR88 inhibition for the highest doses of µOR cDNA transfected. This first series of experiments reveal that GPR88-µOR interactions are critically influenced by partner GPCR expression, and the regular pace of the rightwards shift of µOR signaling dose response when GPR88 amounts increase argues for a 1:1 stoichiometry (or n:n) in this interaction.

Now focusing on µOR mediated recruitment of β-arrestins, increasing µOR expression on GPR88 inhibitory action did not allow full restoration of µOR signaling as for cAMP inhibition. Indeed, no further recruitment was observed over 50 ng of µOR cDNA transfected. Moreover, expressing more GPR88 (50ng of transfected cDNA) led to near complete suppression of β-arrestin recruitment at µOR (Figure 2D). GPR88 inhibitory action on µOR dependent β-arrestin recruitment was thus significantly less sensitive to µOR expression than G-protein dependent signaling, suggesting a different mechanism of interaction. These results are coherent with the hypothesis that we formulate of an intracellular sequestration of β-arrestins by GPR88 (see Figure 6). Indeed, the failure of high doses of µOR to totally rescue β-arr2 recruitment in presence of moderate doses of GPR88 suggests that a pool of β-arrestins remained out of reach of µOR.

Further investigation would be needed to compare the results obtained with µOR to those obtained with other receptors, and assess whether difference in sensitivity of the different GPCR partners of GPR88 may be attributed to difference in the receptive expression of these partners. The above experiments, however, indicate that this is very likely for G-protein dependent signaling when GPR88 comes in close proximity to the partner GPCR (1:1 interaction), but not for β-arrestin recruitment.

The results of previous experiments are now presented in the subsection “GPR88 comes in close physical proximity to opioid receptors and inhibits their signaling” and commented in the Discussion.

3) To be considered as an interesting drug target, as mentioned several times by the authors, it would be necessary to show that the effect reported here on GPR88 can be indeed modulated by drugs. It would be nice if the authors could test at least one of the synthetic ligand reported to active GPR88. Of course, testing an inhibitor would also be a nice addition but as far as I am aware, no such compound is available yet. Alternatively, analyzing the effect of a constitutively active mutant of GPR88, or a loss of function mutation of this receptor would be nice.

We thank the reviewers for this challenging question. Indeed, the fact that we failed to detect any effect of GPR88 activation by Compound 19 on its ability to repress µOR-dependent β-arrestin recruitment in vitro (now Figure 2—figure supplement 2C) appeared somehow disappointing regarding potential druggability of GPR88. To answer the reviewer’s question, we tried to assess the consequences of GPR88 stimulation of µOR-dependent inhibition of cAMP production. Since both GPR88 and µOR recruit Gi proteins when activated, it was not possible to directly distinguish µOR-dependent signaling from GPR88-dependent signaling when the 2 receptors were activated by their respective agonists. We thus subtracted Compound 19-induced cAMP inhibition to the signal measured in presence of both DAMGO and Compound 19, and observed a tendency for an exacerbation of GPR88 blunting effects on µOR-dependent signaling (now Figure 2—figure supplement 2B). We remained very cautious when commenting these results, though, as the method we used to evaluate the inhibitory effects of activated GPR88 on µOR signaling in vitro was rather indirect.

We thus decided to further address the reviewer’s question by evaluating GPR88 druggability in vivo. There is indeed currently no pharmacological antagonist available to inhibit GPR88 activity and evaluate whether its administration would mimic (or not) the effects of *Gpr88* deletion. Thus, as suggested, we administered the synthetic agonist Compound 19 icv to wild-type mice and we assessed whether this administration would mirror the effects of *Gpr88* deletion on morphine-induced locomotor activity and sensitization.

The results of this experiment are now presented in Figure 3 (Figure 3F, panels a and b). Remarkably, Compound 19 reduced drastically morphine-induced locomotor activity in the animals, but had no effect on the amplitude of behavioral sensitization. If the stimulant locomotor effects of morphine are known to involve both µOR (and D1)-mediated recruitment of both G-proteins and β-arrestins, locomotor sensitization was demonstrated to rely essentially on β-arrestin dependent processes. Compound 19 administration thus seems to stimulate the inhibitory effect of GPR88 on µOR-mediated, G-protein-dependent, morphine-induced locomotor stimulation but not on µOR-mediated, β-arrestin-dependent, locomotor sensitization. Interestingly, these results fit with our in vitro observation of a tendency for potentiated the inhibitory effects of GPR88 on µOR-mediated G-protein dependent signaling in presence of its agonist, but no effect of Compound 19 on GPR88’s ability to inhibit µOR-dependent β-arrestin recruitment (Figure 2—figure supplement 2B, C). Whether this would also apply to other GPCR partners of GPR88 (notably partners coupled to Gs or Gq types of G-protein), however, would require further investigation.

These data are now presented in the Results section of the manuscript (subsections “GPR88 comes in close physical proximity to opioid receptors and inhibits their signaling” and “Deletion of Gpr88 gene in mice modifies µOR-mediated responses in a behavior specific manner"), and commented in the Discussion.

4) It become more and more clear that interacting 7TM receptors control each other through direct interaction, and in most cases, the activation of one, prevents the activation of its partner (see the elegant work of M Lohse on this topic using FRET base GPCR sensors (Vilardaga et al., 2008), also supported by many data from the Pin's group (see their recent review on this topic in MCE 2019). As such, GPR88 could inhibit the coupling of a GPCR partner if in an active-like conformation. Can this be tested? This should be mentioned as a possibility at least.

Whether GPR88 needs to be in an active conformation to inhibit the signaling of a GPCR partner is an exciting question. Our previous results showing an absence of effects of GPR88 activation by Compound 19 on its ability to repress µOR-dependent β-arrestin recruitment in vitro (Figure 2—figure supplement 2C) were arguing against this hypothesis. However, the novel data that we obtained by testing the effects of GPR88 activation on its inhibitory action at µOR-mediated G-protein dependent signaling instead suggest that GPR88 influences this signaling when in an active conformation (Figure 2—figure supplement 2B).

Testing the effects of Compound 19 on morphine sensitization in wild-type mice was even more informative regarding this matter. Indeed, Compound 19 inhibited morphine-induced locomotor stimulation (but not β-arrestin-dependent sensitization) in wild-type mice, demonstrating a stimulant effect on the inhibitory action of GPR88 at G-protein-dependent µOR signaling (Figure 3F, panels a and b). This result is fully coherent with the hypothesis that GPR88 is more efficient in suppressing GPCR partner’s G-protein dependent signaling if in an active-like conformation.

These different results are now presented in the Results section (subsections “GPR88 comes in close physical proximity to opioid receptors and inhibits their signaling” and “Deletion of Gpr88 gene in mice modifies µOR-mediated responses in a behavior specific manner") and commented in the Discussion.

5) The pERK signal appears to be very weak, which suggest potential dephosphorylation during sample handling. How were the data normalized?

As concerns ERK phosphorylation assays in HEK cells, the samples were prepared by adding 150 µL of Sample Buffer (125 mM Tris, 20% glycerol, 4% SDS, 10% β-mercaptoethanol, pH=6.8) containing 1X Protease/Phosphatase Inhibitor Cocktail, which stopped DAMGO-induced stimulation and blocked proteases and phosphorylation (and induced cell lysis). They were then frozen at -20°C (this information was missing in the Materials and methods, see now the subsection “Internalization assays”). Freezing may have induced some dephosphorylation in the samples. Normalized values were calculated by dividing the signal from each band of p-ERK and tERK by the GAPDH signal of the corresponding track (now explained in details in the aforementioned subsection). Please note that ERK signal following µOR stimulation is not much intense (see Tao et al., 2017) as compared to the signal triggered by other GPCRs such as the β2-adrenergic receptor (Shenoy et al., 2006).

6) The authors should consider including siRNA experiments to establish that the second wave of Erk activation is arrestin dependent. In the present stage the only thing that can be concluded is that this later phase of signaling is more resistant to PTX. Also, there is a non-replicability at the 30 min time point if data are explained solely by PTX resistant signaling at later phases.

We agree with the reviewers that western blot experiments in Figure 2 fail to provide convincing arguments for an effect of GPR88 co-expression on µOR-induced recruitment of β-arrestins. Based on the reviewers’ comments, we have carefully re-analyzed these data and concluded that evidence for a second peak of phosphorylation following DAMGO stimulation was weak. Indeed, increased ERK phosphorylation at 60 min was mostly driven by one value that we finally decided to exclude (distant from the mean more than 2 fold the standard deviation) (Figure 2A, top panel has been modified accordingly). The late phase of ERK phosphorylation in presence of PTX (but absence of GPR88), however, was robust and well reproduced across five independent experiments. Performing siRNA experiments may have helped understanding the nature of this late response and its potential dependence on β-arrestin recruitment. However, the recent publication by Grundmann et al. (Nature Communications, 2019) suggests that, in the absence of G-protein activation, no recruitment of β-arrestin should be detected. We therefore propose that “This later phase possibly corresponded to incomplete blockade of G protein recruitment by PTX and/or to G protein-independent phosphorylation of ERK”, without referring to β-arrestins. Our conclusions from these experiments are now strictly limited to the G protein-dependent component: “Thus co-expressing GPR88 with µOR represses phosphorylation of the ERK complex through G_i/o_ protein, further demonstrating the inhibitory influence of GPR88 on G-protein dependent µOR signaling” and in the Discussion: “We first confirmed the inhibitory effect of GPR88 expression on G-protein dependent signaling of µOR, by evidencing a suppression of G_i/o_ protein-dependent µOR-induced phosphorylation of ERK in presence of GPR88”.